# faSavageHutterFOAM 1.0: Depth-integrated simulation of dense snow avalanches on natural terrain with OpenFOAM

Matthias Rauter[1,2,3], Andreas Kofler[2], Andreas Huber[4], and Wolfgang Fellin[1]

[1]University of Innsbruck, Institute of Infrastructure, Division of Geotechnical and Tunnel Engineering
[2]Department of Natural Hazards, Austrian Research Centre for Forests (BFW), Innsbruck, Austria
[3]Norwegian Geotechnical Institute, Oslo, Norway
[4]University of Innsbruck, Institute of Infrastructure, Division of Hydraulic Engineering

**Abstract.**

Numerical models for dense snow avalanches have become a central part in hazard zone mapping and mitigation. Several commercial and free applications, which are used on a regular basis, implement such models. In this study we present a tool based on the open-source toolkit OpenFOAM® as an alternative to the established solutions. The proposed tool implements a depth-integrated shallow flow model in accordance to current practice. The solver combines advantages of the extensive OpenFOAM infrastructure with popular models from the avalanche community. OpenFOAM allows assembling of custom physical models with build-in primitives and implements the numerical solution at a high level. OpenFOAM supports an extendable solver structure, making the tool well-suited for future developments and rapid prototyping. We introduce the basic solver, implementing an incompressible, single-phase model for natural terrain, including entrainment. The respective workflow, consisting of meshing, pre-processing, numerical solution and post-processing, is presented. We demonstrate data transfer from and to a geographic information system (GIS), to allow a simple application in practice. The tool-chain is based entirely on open-source applications and libraries and can be easily customized and extended. Simulation results for a well documented avalanche event are presented and compared to previous numerical studies and historical data.

## 1 Introduction

Numerical avalanche modelling has become an important and well-accepted ingredient to hazard zone mapping. All popular tools rely on depth-integrated flow models (Pudasaini and Hutter, 2007) and only a few academic exceptions are known (Domnik et al., 2013; Kröner, 2013; von Boetticher et al., 2016, 2017; Barker and Gray, 2017). Depth-integrated flow models, widely known as Shallow Water Equations, have a long tradition in hydraulic modelling (e.g., Vreugdenhil, 1994), dating back to Barré de Saint-Venant (1871). This approach is commonly applied in academia and practice because it reduces the computational effort to a level, where physical simulations of realistic flows are feasible. The first application to gravitational mass flows is attributed to Grigorian et al. (1967), the first formal derivation and analysis of the underlying model to Savage and Hutter (1989, 1991). Since then, the mechanical model has been continuously improved and extended to e.g., simple, two-dimensional surfaces (Greve et al., 1994), complex, shallow surfaces (Gray et al., 1999) or curved and twisted flow paths (Pudasaini et al., 2005a, b). Finally, respective models have been adapted to natural, i.e. arbitrary but mildly curved terrain,

making simulations of real case avalanches possible. The limitation to mildly curved terrain requires the flow thickness to be small in relation to the curvature radius of the surface. Denlinger and Iverson (2004) proposed a model embedded in an ordinary Cartesian coordinate system as an alternative to the complex curvilinear coordinate system used by Savage and Hutter (1989, 1991). Bouchut and Westdickenberg (2004), Hergarten and Robl (2015) and recently Rauter and Tuković (2018) follow a similar approach. Christen et al. (2010) apply a non-orthogonal local coordinate system (Fischer et al., 2012), however, without incorporating the respective correction terms (Hergarten and Robl, 2015). A Lagrangian solution, which has some advantages for natural terrain, has been presented by Hungr (1995) and later on by Sampl and Zwinger (2004) and Sampl and Granig (2009).

Beside improvement of the underlying mechanical model, various physical processes have been added to governing equations, such as multiple phases (e.g., Pudasaini, 2012; Kowalski and McElwaine, 2013; Iverson and George, 2016), entrainment (e.g., Issler, 2014), improved basal friction relations (e.g., Voellmy, 1955; Norem et al., 1987; Pouliquen and Forterre, 2002; Bartelt et al., 2006; Issler and Gauer, 2008; Baker et al., 2016; Rauter et al., 2016), (for a review see Ancey, 2007), compressibility (e.g., Iverson and George, 2014; Bartelt et al., 2015) or thermodynamic processes (e.g., Vera Valero et al., 2015).

In this work, we strictly distinguish between mechanical model and process models. The mechanical model consists of basic conservation equations and their reformulation, e.g. in terms of depth-integration. Process models, on the other hand, describe closure of governing equations with e.g. constitutive models. The combination of the mechanical model and all closures is called flow model or physical model throughout this work.

There are several numerical methods to solve the respective mathematical equations. Basically, most methods can be classified as finite difference method (e.g., Wang et al., 2004), finite element method (e.g., Hanert et al., 2005), finite volume method (e.g., Christen et al., 2010) or as Lagrangian particle method (e.g., Sampl and Granig, 2009). Specialised differencing schemes (e.g., upwind, TVD, NVD) prevent oscillations (e.g., Jasak et al., 1999).

Shallow granular flow models have been carefully validated over the last few decades. This includes back-calculations of small scale experiments (for a review see Pudasaini and Hutter, 2007), large scale experiments (e.g., Christen et al., 2010), historic snow avalanches (e.g, Fischer et al., 2015) and rock avalanches (e.g., Mergili et al., 2017). Shallow flow models have various weaknesses, such as the limitation to mildly curved terrain or the missing resolution in surface normal direction. However, they have proven to be a good trade-off between accuracy and computing time and thus useful for many applications.

Shallow flow models gained popularity through commercial software packages: DAN (Hungr, 1995), SamosAT (Sampl and Zwinger, 2004), FLATModel (Medina et al., 2008) and RAMMS (Christen et al., 2010) implement such models and are used regularly in practice. Open-source alternatives include TITAN2D (Pitman et al., 2003; Patra et al., 2005), r.avaflow (Mergili et al., 2012, 2017) and an extension to the CFD-toolkit (computation fluid dynamics) GERRIS (Hergarten and Robl, 2015). From an academic viewpoint, open-source applications have various advantages over their commercial counterparts; e.g. users can view and modify the source code to gain a better understanding of the software and adapt the flow model without re-implementing basic models and numerical methods from scratch.

Geographic Information Systems (GIS) are commonly applied in hazard zone mapping. Therefore numerical simulation tools are usually incorporated or linked to these systems to streamline the respective workflow. GIS allows user friendly data input, post-processing and production of publication quality maps.

Recently, Rauter and Tuković (2018) proposed a shallow granular flow model, expressed in terms of surface partial differential equations (Deckelnick et al., 2005; Tuković and Jasak, 2012) and presented an open-source implementation based on the CFD-toolkit OpenFOAM® (OpenCFD Ltd., 2004). The underlying mechanical model is widely similar to the classic Savage and Hutter (1989, 1991) model and its derivations.

One particular advantage of an OpenFOAM solver is the well-designed, object-oriented source code. This makes the code cleaner than comparable solutions as it hides implementation details, such as numerical schemes, I/O or inter-process communication, behind well defined interfaces. The top-level solver mimics the tensorial notation of partial differential equations and specific implementations of e.g., interpolation schemes, are exchangeable without changing the top-level source code. This enables separation of physical models and numerical solution, which allows a streamlined interdisciplinary development process. Process models, e.g. entrainment and basal friction can be incorporated similarly, keeping the source code clean and easy to extend.

The OpenFOAM solver, presented in here, implements an incompressible single-phase model including various basal friction and entrainment closures. The solver is called *faSavageHutterFoam*, indicating that the underlying mechanical model is similar to the one of Savage and Hutter (1989, 1991), however with exchangeable closure models. This model is, to some extent, suitable for dense snow avalanches and constitutes the baseline for complex flow models, as employed by e.g. Bartelt et al. (2015) or Mergili et al. (2017). Moreover, the underlying method has been developed to simplify coupling with three-dimensional ambient flows (Tuković and Jasak, 2012; Marschall et al., 2014; Dieter-Kissling et al., 2015a, b; Pesci et al., 2015), which enables development of models for mixed snow avalanches (e.g., Sampl and Zwinger, 2004) and turbidity currents (e.g., Huang et al., 2005).

The purpose of this paper is to present the capability of the new OpenFOAM solver and the Rauter and Tuković (2018) model. The solver is evaluated and validated for snow avalanches on natural terrain. We present the basic flow model, as well as methods and tools to incorporate natural terrain and GIS data in OpenFOAM simulations. Also export of OpenFOAM results to a GIS for post-processing and visualisation is demonstrated. Results for a well documented avalanche event are presented and compared to historical records and results of SamosAT. All underlying source code (except SamosAT) and data are available for free to encourage reproduction, improvement and cross validation.

## 2 Method

### 2.1 Flow model

Historically, shallow granular flow models have been set up in surface aligned, curvilinear coordinates, leading to a two-dimensional system of partial differential equations (e.g., Savage and Hutter, 1989, 1991). Rauter and Tuković (2018) follow a different approach (see also, e.g., Denlinger and Iverson, 2004; Bouchut and Westdickenberg, 2004; Hergarten and Robl,

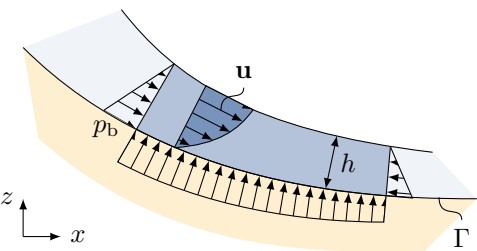

**Figure 1.** Definition of velocity $\mathbf{u}$, flow thickness $h$ and basal pressure $p_\mathrm{b}$ on a control volume. A hydrostatic and linear pressure distribution is assumed. The shape of the velocity profile is commonly ignored in governing equations (Baker et al., 2016). Flow thickness $h$ is measured normal on the basal surface $\Gamma$. The curvature radius of the surface $\Gamma$ is assumed to be much bigger than the flow thickness.

2015) and formulate the mechanical model in terms of surface partial differential equations (SPDEs, e.g., Deckelnick et al., 2005). Respective SPDEs are defined on a surface $\Gamma$, embedded in three-dimensional space, which represents the mountain topography. This approach, popular in the thin liquid film community (e.g., Craster and Matar, 2009), avoids transformations into the surface aligned coordinate system and thus complex metric tensors. Considering the relative shallowness of the avalanche, it can be treated as thin layer flowing along the mountain surface. The governing equations describe the motion of the avalanche in three-dimensional space along this surface. Consequently, velocity is a three-dimensional vector field and contains all information on flow direction and respective effects, such as centrifugal forces. Resulting SPDEs can be solved with various methods, e.g. finite element method (e.g., Olshanskii et al., 2009) or finite area method, a modified finite volume method (Tuković and Jasak, 2012).

### 2.1.1 Mechanical model

A basic shallow granular flow model can be written in terms of surface partial differential equations as[1]

$$\frac{\partial h}{\partial t} + \boldsymbol{\nabla} \cdot (h\,\overline{\mathbf{u}}) = \frac{\dot{q}}{\rho}, \tag{1}$$

$$\frac{\partial (h\,\overline{\mathbf{u}})}{\partial t} + \boldsymbol{\nabla}_\mathrm{s} \cdot (h\,\overline{\mathbf{u}}\,\overline{\mathbf{u}}) = -\frac{1}{\rho}\boldsymbol{\tau}_\mathrm{b} + h\,\mathbf{g}_\mathrm{s} - \frac{1}{2\rho}\boldsymbol{\nabla}_\mathrm{s}\,(h\,p_\mathrm{b}), \tag{2}$$

$$\boldsymbol{\nabla}_\mathrm{n} \cdot (h\,\overline{\mathbf{u}}\,\overline{\mathbf{u}}) = h\,\mathbf{g}_\mathrm{n} - \frac{1}{2\rho}\boldsymbol{\nabla}_\mathrm{n}\,(h\,p_\mathrm{b}) - \frac{1}{\rho}\mathbf{n}_\mathrm{b}\,p_\mathrm{b}. \tag{3}$$

Equations (1) to (3) are equivalent to a Savage-Hutter like system, consistently extended to complex but mildly curved terrain and entrainment. The notation as SPDE makes extension to complex terrain straight-forward and implementation into SPDE environments, e.g. OpenFOAM, possible. A formal derivation is given by Rauter and Tuković (2018). Here, we aim to deliver a short and descriptive introduction.

Equation (1) represents the depth-integrated continuity equation, Eq. (2) the surface tangential momentum conservation equation and Eq. (3) its surface normal counter part, defined in all points $\mathbf{x}_\mathrm{b}$ on the surface $\Gamma \subset \mathbb{R}^3$, representing the mountain

---

[1]Multiplications between vectors represent the outer product $\mathbf{u}\,\mathbf{v} = \mathbf{u} \otimes \mathbf{v}$

surface. The time is denoted as $t$. The unknown fields are the surface normal flow thickness $h(\mathbf{x}_\mathrm{b})$ (see Fig. 1), the depth-averaged flow velocity $\overline{\mathbf{u}}(\mathbf{x}_\mathrm{b}) \in \mathbb{R}^3$, defined as

$$\overline{\mathbf{u}}(\mathbf{x}_\mathrm{b}) = \frac{1}{h(\mathbf{x}_\mathrm{b})} \int\limits_0^{h(\mathbf{x}_\mathrm{b})} \mathbf{u}(\mathbf{x}_\mathrm{b} - \mathbf{n}_\mathrm{b}\, z')\, \mathrm{d}z', \qquad (4)$$

and the basal pressure $p_\mathrm{b}(\mathbf{x}_\mathrm{b})$. The density $\rho$ is assumed to be constant. Note that the earth pressure theory (e.g., Savage and Hutter, 1989, 1991) has been replaced with the hydrostatic pressure assumption, as in most practical applications (e.g., Christen et al., 2010). Moreover, Eqs. (1) to (3) are written in conservative form. Therefore, there is no entrainment term in Eq. (2), which would show up in a non-conservative formulation. The first terms in Eqs. (1) and (2) represent the temporal derivative, i.e. the local change of mass and momentum, respectively. The second terms in Eqs. (1) and (2) are the respective advection terms. The right hand side of Eq. (1) represents mass growth due to entrainment. The first, second and third terms on the right hand side of Eq. (2) represent surface tangential components of basal friction, gravitational acceleration and lateral pressure gradient, respectively. The surface normal components of these terms appear in the surface normal momentum conservation equation (3). This equation is used to calculate the basal pressure, represented by the last term.

In the framework of SPDEs, the normal vector field $\mathbf{n}_\mathrm{b}(\mathbf{x}_\mathrm{b}) \in \mathbb{R}^3$ of the surface $\Gamma$ is sufficient to describe all major curvature effects. This is realised by calculating all contributions to conservation equations in the global coordinate system and projecting results on the surface and the surface normal vector, respectively. These projections are explained in detail in appendix A. Surface tangential and normal components contribute to local acceleration and basal pressure, respectively. This follows from the assumption that movement is constrained in surface normal direction, which is enforced by a mechanical force, namely the basal pressure. The gravitational acceleration $\mathbf{g}$, e.g., is split into a surface tangential component,

$$\mathbf{g}_\mathrm{s} = (\mathbf{I} - \mathbf{n}_\mathrm{b}\,\mathbf{n}_\mathrm{b}) \cdot \mathbf{g}, \qquad (5)$$

and surface normal component,

$$\mathbf{g}_\mathrm{n} = (\mathbf{n}_\mathrm{b}\,\mathbf{n}_\mathrm{b}) \cdot \mathbf{g}. \qquad (6)$$

The gradient operator $\boldsymbol{\nabla}$ denotes the three-dimensional derivative along the surface (Deckelnick et al., 2005). If the responding result is a three-dimensional vector field (e.g. gradient of a scalar field or divergence of a tensor field), it can be split, similar to the gravitational acceleration, into a surface tangential component,

$$\boldsymbol{\nabla}_\mathrm{s} = (\mathbf{I} - \mathbf{n}_\mathrm{b}\,\mathbf{n}_\mathrm{b}) \cdot \boldsymbol{\nabla}, \qquad (7)$$

and surface normal component,

$$\boldsymbol{\nabla}_\mathrm{n} = (\mathbf{n}_\mathrm{b}\,\mathbf{n}_\mathrm{b}) \cdot \boldsymbol{\nabla}. \qquad (8)$$

For simply curved surfaces, the given relation matches the model of Greve et al. (1994), as shown by Rauter and Tuković (2018).

### 2.1.2 Process models

There are various user-selectable models, describing basal friction $\boldsymbol{\tau}_{\mathrm{b}}(\mathbf{x}_{\mathrm{b}})$ and entrainment rate $\dot{q}(\mathbf{x}_{\mathrm{b}})$, to close the system of equations. To reassemble the traditional model (often called Voellmy or Voellmy-Salm model, Christen et al., 2010), as applied by e.g. Fischer et al. (2015), the basal friction is described following Voellmy (1955),

$$5 \quad \boldsymbol{\tau}_{\mathrm{b}} = \mu \, p_{\mathrm{b}} \frac{\overline{\mathbf{u}}}{|\overline{\mathbf{u}}| + u_0} + \frac{\rho \, g}{\xi} \, |\overline{\mathbf{u}}| \, \overline{\mathbf{u}}. \tag{9}$$

Therein $\mu$ and $\xi$ are constant parameters, although they may depend on avalanche size and surface roughness (Salm et al., 1990) or flow regime (Köhler et al., 2016). The small value $u_0$ ($10^{-7}\,\mathrm{m\,s^{-1}}$ in here) avoids divisions by zero and regularizes the relation near still-stand, where the original function is discontinuous. This regularisation, combined with the employed time integration scheme (implicit three-level second-order, Ferziger and Peric, 2002), leads to a well-defined behaviour in the runout

10 zone, where the velocity is nearly zero (Rauter and Tuković, 2018). This allows the avalanche to reach very low velocities in the runout zone, which are lower than the tolerance of the solver and thus virtually zero. For characteristic avalanche velocities, i.e. $|\overline{\mathbf{u}}| > 100 \, u_0$, this value has no relevant effect on the dynamic behaviour. Previously, this issue has been addressed with operator splitting and explicit stress reduction (e.g., Mangeney-Castelnau et al., 2003; Zhai et al., 2015; Mergili et al., 2017), which is not required in the proposed scheme.

15 The entrainment rate is calculated, based on an empirical erosive entrainment model, as

$$\dot{q} = \begin{cases} \dfrac{\boldsymbol{\tau}_{\mathrm{b}} \cdot \overline{\mathbf{u}}}{e_{\mathrm{b}}} & \text{for} \quad h_{\mathrm{msc}} > 0, \\ 0 & \text{for} \quad h_{\mathrm{msc}} = 0, \end{cases} \tag{10}$$

where $e_{\mathrm{b}}$ is the specific erosion energy (Fischer et al., 2015). Entrainment is restricted by the available mountain snow cover thickness $h_{\mathrm{msc}}$. The initial mountain snow cover thickness is calculated following Fischer et al. (2015), using a linear approach,

$$20 \quad h_{\mathrm{msc}}(z) = \left( H_{\mathrm{msc}}(z_0) + \frac{\partial H_{\mathrm{msc}}}{\partial z} \, (z - z_0) \right) \cos(\zeta), \tag{11}$$

where $z$ is the surface elevation (corresponding to the vertical coordinate in the numerical model), and $z_0$ the elevation of a reference station, which has to be provided by the user, alongside with the base value $H_{\mathrm{msc}}(z_0)$ and the growth rate $\frac{\partial H_{\mathrm{msc}}}{\partial z}$. $\zeta$ is the angle between the gravitational acceleration and the surface normal vector. Its further evolution is described by the conservation equation

$$25 \quad \frac{\partial h_{\mathrm{msc}}}{\partial t} = -\frac{\dot{q}}{\rho}. \tag{12}$$

Undershoots, i.e. $h_{\mathrm{msc}} < 0$, are prevented with a regularisation similar to Eq. (9). This can be realised by multiplying the entrainment rate $\dot{q}$ with $\frac{h_{\mathrm{msc}}}{h_{\mathrm{msc}} + h_0}$, where $h_0$ is a small value, similar to $u_0$.

### 2.1.3 Numerical solution

The governing equations are solved with an implicit, conservative, finite area method (Rauter and Tuković, 2018), using the

30 respective OpenFOAM library (Tuković and Jasak, 2012). The finite area method is similar to the well-known finite volume

method (e.g. Jasak, 1996), however with appropriate differential operators for SPDEs, Eqs. (7) and (8). We apply first- (upwind scheme) and second-order accurate spatial differencing schemes. First-order schemes converge slower in terms of mesh refinement due to their high numerical diffusivity. However, they effectively prevent oscillations and increase the stability of the solver. Oscillations in second-order accurate simulations are prevented with a normalised variable diagram (NVD) scheme for unstructured meshes, known as Gamma scheme (Jasak et al., 1999). NVD schemes blend upwind and a higher order scheme to combine advantages of both methods.

As mentioned before, OpenFOAM utilises capabilities of C++ to make top-level source code appear similar to the tensor notation of partial differential equations. The conservation equation (1), e.g., can be solved with the following lines of code using OpenFOAM:

```
faScalarMatrix hEqn
(
    fam::ddt(h)
  + fam::div(phis, h)
 ==
    dqdt/rho
);
hEqn.solve();
```

`phis` is the velocity edge field (see Rauter and Tuković, 2018, for details) and `dqdt` the source term incorporating entrainment. Momentum conservation equations (2) and (3) look similar (see Rauter and Tuković, 2018) and conservation equations for arbitrary fields (e.g. random kinetic energy, Bartelt et al., 2015) can be added with the same syntax.

## 2.2 Simulation evaluation

We use an established implementation of the same flow model, SamosAT (version 2017_07_05) (Sampl and Zwinger, 2004; Sampl and Granig, 2009), for comparison. The main difference between SamosAT and the presented OpenFOAM solver is the solution method. SamosAT solves similar governing equations, slightly adapted to fit into the respective framework, with smoothed-particle hydrodynamics (SPH). This approach follows a Lagrangian description, making handling of complex terrain simpler (Sampl and Zwinger, 2004). Therefore, SamosAT provides an excellent reference to validate avalanche models for complex terrain. The second term on the right hand side of Eq. (3) was deactivated in OpenFOAM computations to reassemble the mechanical model as implemented in SamosAT. This term is usually small and can be safely neglected (Rauter and Tuković, 2018). However, it is shown in equations to preserve the similarity between Eqs. (2) and (3).

We compare simulations using the $1\,\mathrm{kPa}$-isoline of the dynamic peak pressure, defined as

$$p_{\mathrm{dyn}}(\mathbf{x}_{\mathrm{b}}) = \max_{t}\left(\rho\,|\overline{\mathbf{u}}(\mathbf{x}_{\mathrm{b}},t)|^2\right). \tag{13}$$

Definitions of hazard zones are based on this threshold in many European countries (Jóhannesson et al., 2009) and therefore often used for evaluation of respective models (e.g., Fischer et al., 2015; Rauter et al., 2016).

In addition to the comparison with a reference implementation, we present a comparison with historical records from a catastrophic event. A common method to document avalanches is the delineation of deposition. This information is also available for the presented case study. Deposition processes are not explicitly included in the flow model due to depth-integration. However, the general form and size of the deposition should be reproduced by the model to be useful for hazard zone mapping. This is problematic in some implementations, e.g., SamosAT, due to missing regularisation of the friction term, but possible with the proposed method.

We apply model parameters ($\mu$, $\xi$, $e_\mathrm{b}$) optimised for SamosAT (Fischer et al., 2015) and the comparison is conducted on a qualitative level.

Finally, we evaluate OpenFOAM simulations with regard to convergence during mesh refinement to give a quantitative estimation of numerical uncertainties as recommended by Roache (1997). The numerical solution should converge to the unknown analytical solution with increasing grid resolution and the numerical uncertainty should decay with the order of the applied method. Richardson extrapolation allows to estimate the numerical uncertainty, using results of three different meshes. This way, the expected convergence can be verified and the numerical uncertainty quantified.

## 2.3  Simulation setup

The precondition to conduct simulations in OpenFOAM is a mesh, describing the geometry of the problem. For SPDEs, e.g. shallow flow models, a surface mesh, matching the slope topography, is sufficient and no volume mesh is required. In practice, however, three-dimensional meshing tools can be used to create a volume mesh, the boundary of which can be used as surface mesh.

Topography is usually available as digital elevation model (DEM) in GIS formats, yielding elevation on a regular two-dimensional grid. The relevant part of the topography is re-sampled with cubic splines, triangulated and stored as STL file (e.g., Kai et al., 1997) to prepare it for meshing. We chose the meshing application cfMesh (Juretić, 2015), because of its good integration in OpenFOAM and its clean boundary meshes. cfMesh requires a closed triangulated surface to create a volume mesh. This is the case for all general purpose meshing tools and cfMesh can be easily replaced in our tool-chain, for example with Netgen (Schöberl, 1997) (see Rauter and Tuković, 2018, for an application). Various other meshing tools can be applied and OpenFOAM provides a large range of mesh conversion tools. The closed surface can be assembled from a triangulation of the mountain surface, sidewalls and the respective top boundary. The resulting surface and volume mesh are presented in Figs. 3b and 3c. Refinement near the mountain surface reduces the amount of required volume cells, while keeping the number of surface cells high. The resulting mesh is also valid for three-dimensional simulations with e.g. Navier Stokes Equations, as conducted by e.g. Sampl and Zwinger (2004); Dutykh et al. (2011); Kröner (2013); von Boetticher et al. (2016, 2017); Huang et al. (2005). The boundary mesh, describing the mountain surface, is shown in Fig. 3d. The shallow flow model is solved on this surface mesh. We used polygonal-dominated (volumetric polyhedral-dominated, respectively) meshes for simulations because of stability and accuracy reasons (Juretić, 2005). Triangular (volumetric tetrahedral, respectively) meshes have been evaluated as well. However, second-order accurate simulations on triangular meshes failed, while first-order accurate simulations are virtually identical to the respective simulations on polygonal-dominated meshes.

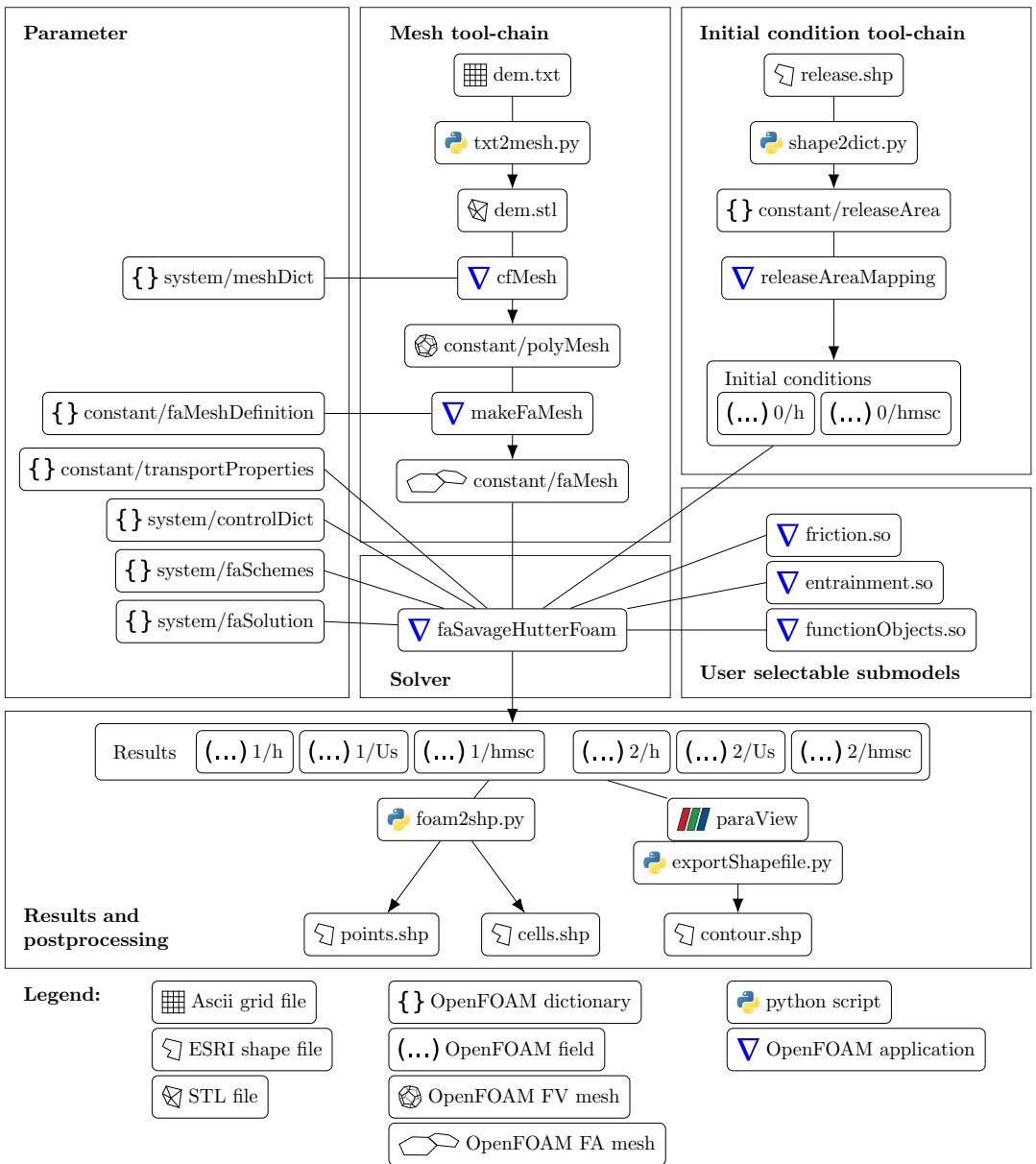

**Figure 2.** Simulation setup and tool-chain. The tool-chain consists exclusively of open-source applications. Individual applications and process models can be replaced with custom ones. Parameters for python scripts are provided via command line interface. Parameters for OpenFOAM applications are provided through OpenFOAM dictionaries. Domain decomposition and reconstruction, which is handled by separate applications, is not shown. OpenFOAM reads initial conditions from the folder "0" and writes results to folders, named after the respective timestep ("1", "2", etc.). Details on OpenFOAM formats can be found in OpenCFD Ltd. (2004).

The release area, acting as initial condition, is provided as a polygon in ESRI shapefile format (ESRI, 1998). To find all surface cells within the given polygon, the Hormann and Agathos (2001) algorithm as implemented in OpenFOAM is applied. The mountain snow cover $h_{\mathrm{msc}}$ of respective cells is then transferred to the flow thickness $h$ to create a suitable initial condition. The release area for our case study, taken from Fischer et al. (2015), is shown in Fig. 7a as polygon and as set of surface cells in Fig. 3d.

The solver reads the surface mesh and initial conditions, as well as physical models, numerical schemes and constants to initialise the simulation (see Fig. 2). The respective entries can be found in the designated locations, according to usual practice in OpenFOAM (OpenCFD Ltd., 2004). The solver can run on multiple processors using domain decomposition (Weller et al., 1998) and message passing interface (MPI).

User defined friction and entrainment models can be loaded at run-time, meaning that the user does not have to recompile the solver to add a custom friction or entrainment model. The same is the case for general purpose functions which are triggered at the end of every time-step. Here, we used this interface to calculate and record the dynamic peak pressure at run-time, without the necessity to save multiple time-steps or to change solver source code. Similar functions can be used to check mass, momentum or energy conservation, record specific data (e.g. time-line at a certain point) or to manipulate fields during run-time, e.g. to trigger secondary slabs.

Simulation results are written to hard disk in the usual OpenFOAM file format (OpenCFD Ltd., 2004) for post-processing, evaluation and simulation restart. The simulation setup, all involved applications and all intermediate and final files are presented in Fig. 2. The tool-chain is modularly assembled from various open-source applications. Single modules, such as mesher, solver or friction model, can be easily replaced.

## 2.4 Post-processing and visualisation

Post-processing and visualisation of OpenFOAM simulations is commonly performed using ParaView® (Ahrens et al., 2005; Ayachit, 2015) (see Figs. 3, 4 and 5). ParaView is an open-source data analysis and visualisation application. It can read and visualise OpenFOAM files and they can be used for further operations, such as the calculation of contour lines. To integrate GIS applications in post-processing, results can be exported to common GIS file formats. Contour lines can be exported to ESRI shapefile format with a custom python extension based on the library pyshp (Figs. 6c, 7 and 9). Alternatively, individual cells and respective field values, can be exported as polygons (Fig. 6a) or points (Fig. 6b) to ESRI shapefiles.

To generate regular raster files, the unstructured OpenFOAM mesh and associated fields have to be mapped to a structured Cartesian grid (Figs. 6d and 9). These and other approaches allow an almost seamless integration into general purpose GIS applications, as shown in the following case study. Here, we utilise foam-extend 4.0 with a custom solver, python 2.7.12 with numpy 1.11.0, scipy 0.17.0 and pyshp 1.2.3 for shapefile export, ParaView 5.0.1 and QGis 2.8.6.

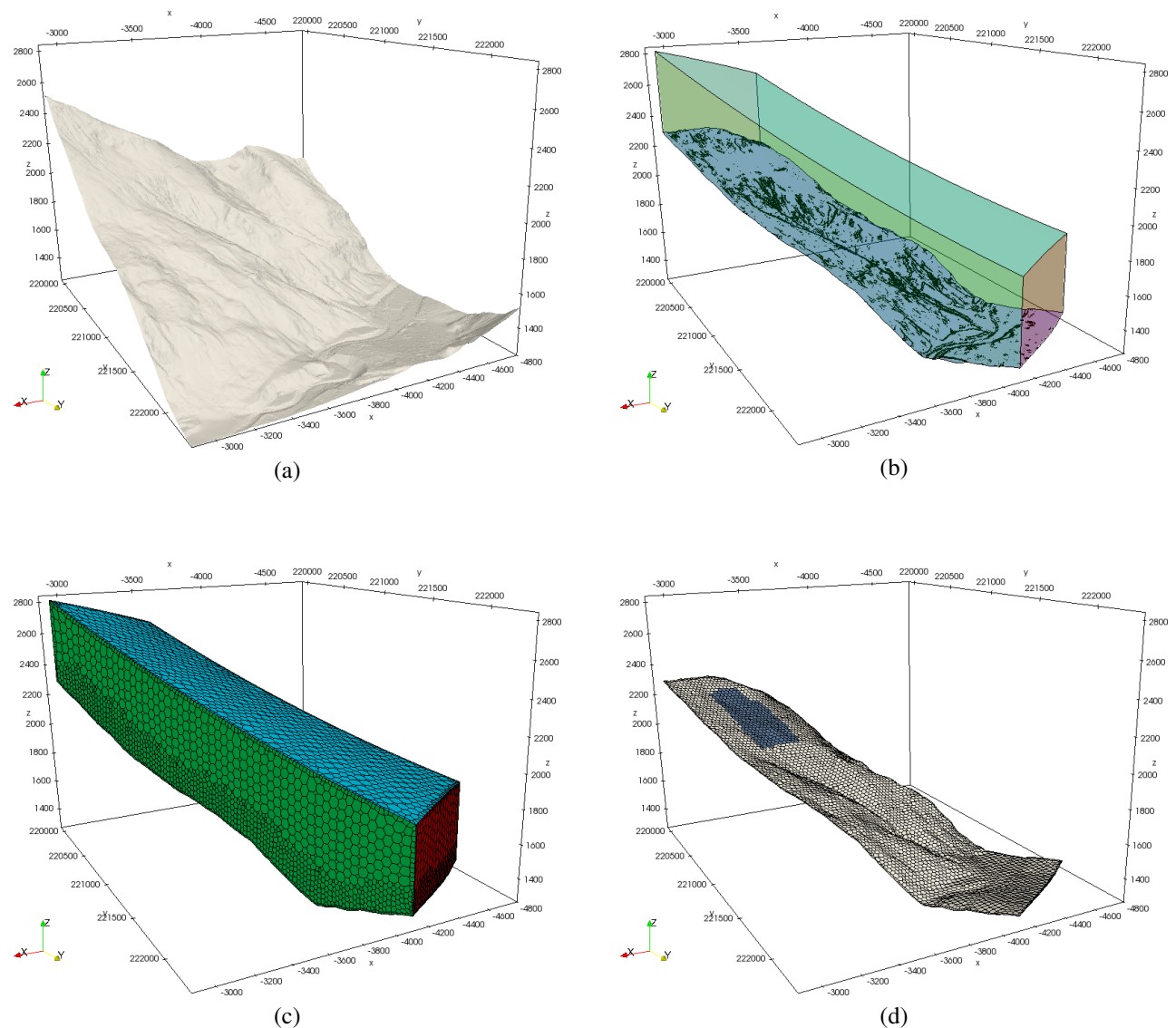

**Figure 3.** Meshing tool-chain: The terrain data is usually available as raster data (a). Triangulation of the relevant area and adding walls and a top boundary yields a closed triangulated surface (b, sharp edges are highlighted black). This surface can be processed by most meshing tools, here we apply cfMesh to get a polyhedral-dominated finite volume mesh (c). The bottom boundary surface of the finite volume mesh builds the foundation for the finite area mesh used for simulations (d). Note that we show a very coarse mesh for the sake of visibility of edges. Terrain data: Amt der Tiroler Landesregierung (AdTLR). EPSG: 31254.

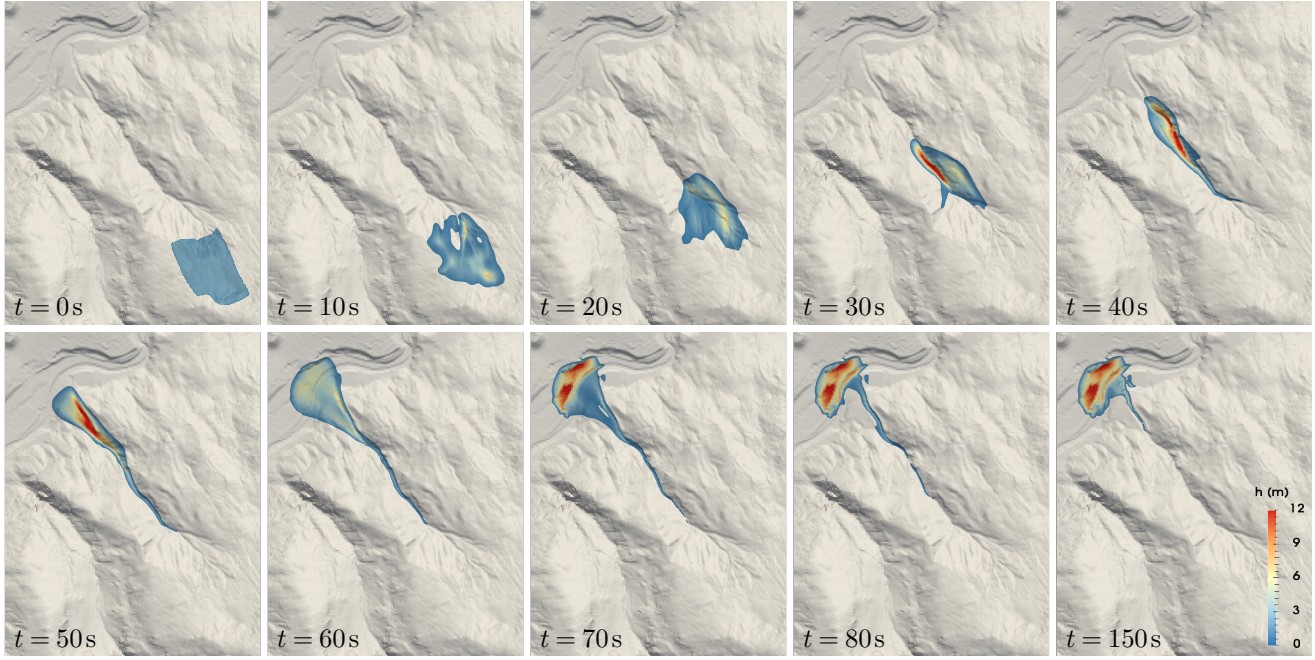

**Figure 4.** Time series of an OpenFOAM simulation with mean cell size $\Delta = 7.45\,\text{m}$ and first-order interpolations in ParaView. The colour scale represents flow thickness, which is clipped at $0.5\,\text{m}$. Terrain data: AdTLR.

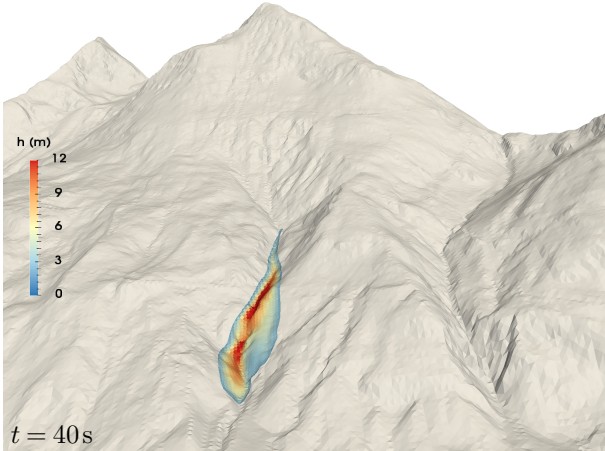

**Figure 5.** Perspective view on the OpenFOAM simulation with mean cell size $\Delta = 7.45\,\text{m}$ and first-order interpolations in ParaView. The colour scale represents flow thickness, which is clipped at $0.5\,\text{m}$. Terrain data: AdTLR.

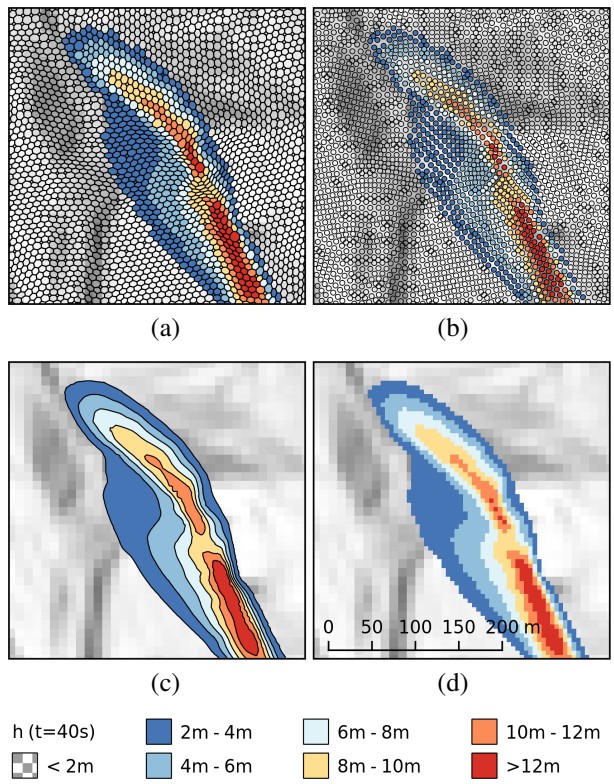

(a) (b)

(c) (d)

| h (t=40s) | 2m - 4m | 6m - 8m | 10m - 12m |
| < 2m | 4m - 6m | 8m - 10m | >12m |

**Figure 6.** The flow thickness field $h$ at time $t = 40\,\text{s}$ for a simulation with mean cell size $\Delta = 7.45\,\text{m}$, first-order interpolations. The figure shows four methods to export and analyse results in GIS: Export of cells as polygons (a). Export of cell centres as points (b). Export of contour lines as polygons (c). Remapping of the unstructured finite area mesh to regular raster (d). The raster has been created by converting point data to raster file in QGIS. The resolution of the DEM is $10\,\text{m}$, results have been mapped to a $5\,\text{m}$ grid. Terrain data: AdTLR.

## 3 Case study

In this work we focus on the Wolfsgruben avalanche. The event from the $13^{\text{th}}$ March 1988, when the avalanche struck inhabited areas has been repeatedly used as benchmark for avalanche simulations, latest by Fischer et al. (2015). We chose this example because the respective data is freely available, making reproduction and cross validation possible.

5  The mountain snow cover thickness for the specific event can be described with parameters $H_{\text{msc}}(z_0) = 1.61\,\text{m}$, $z_0 = 1\,289\,\text{m}$, $\frac{\partial H_{\text{msc}}}{\partial z} = 8 \cdot 10^{-4}$. Physical parameters to reassemble the runout properly are $\mu = 0.26$, $\xi = 8\,650\,\text{m}\,\text{s}^{-2}$ and $e_{\text{b}} = 11\,500\,\text{J}\,\text{kg}^{-1}$. These parameters have been optimised in a previous study using SamosAT (Fischer et al., 2015).

Numerical parameters for OpenFOAM (see Rauter and Tuković, 2018) have been chosen such, that they do not influence the results, while keeping the solver as stable as possible. The appropriate mesh resolution for OpenFOAM has been identified

10 using a mesh refinement study, which is presented alongside the results. The simulation duration has been set to $150\,\text{s}$. This duration is sufficient to reach still-stand (i.e. velocity lower than the solver tolerance, $|\overline{\mathbf{u}}| < 10^{-5}\,\text{m}\,\text{s}^{-1}$) in the runout zone

**Table 1.** Mesh size, runout, error estimation and execution time for different OpenFOAM simulations. Base cell size and refinements refer to parameters of cfMesh.

| interpolation | base cell size | refinements | number of cells | mean cell size | runout | num. uncertainty | exec. time |
|---|---|---|---|---|---|---|---|
| $1^{st}$-order | 40 m | 2 | 40899 | 7.45 m | 2145 m | | 173 s |
| $1^{st}$-order | 30 m | 2 | 72166 | 5.61 m | 2137 m | 46 m | 396 s |
| $1^{st}$-order | 20 m | 2 | 161364 | 3.75 m | 2112 m | 6 m | 1261 s |
| $1^{st}$-order | 15 m | 2 | 285892 | 2.82 m | 2107 m | | 3051 s |
| $2^{nd}$-order | 40 m | 2 | 40899 | 7.45 m | 2156 m | | 353 s |
| $2^{nd}$-order | 30 m | 2 | 72166 | 5.61 m | 2142 m | 66 m | 810 s |
| $2^{nd}$-order | 20 m | 2 | 161364 | 3.75 m | 2109 m | 2 m | 2737 s |
| $2^{nd}$-order | 15 m | 2 | 285892 | 2.82 m | 2107 m | | 6952 s |

and thus virtually unchanging deposition. We decomposed the simulation domain into four parts for OpenFOAM and all simulations have been conducted on a Quadcore Intel Core i7-7700K @ 4.20 GHz and 32 GB DDR4 Ram @ 2.667 GHz.

SamosAT utilises a grid with 5 m resolution and we follow recommendations in terms of appropriate particle numbers and other numerical parameters. The interpolation method has been varied between interpolation on grid (SPH-mode 0) and interpolation on particles (SPH-mode 1) to get an insight into the numerical uncertainty.

ParaView renderings are presented in Fig. 4 for multiple time-steps, showing the dynamic behaviour of the avalanche. A perspective ParaView rendering is shown in Fig. 5. The avalanche follows the narrow channel directly beneath the release area. Small portions of the avalanche overflow the left and right humps in some simulations, which can be seen in the peak dynamic pressure, Fig. 7.

The results at time-step $t = 40$ s have been exported to QGIS using various methods, see Fig. 6. Affected areas (i.e. 1 kPa-isolines), as predicted by OpenFOAM and SamosAT are shown in Fig. 7. Variations due to different interpolation schemes are shown for both implementations, to give an insight into the numerical uncertainty.

The influence of the mesh resolution on the affected area is shown in Fig. 8 for the OpenFOAM solver. Respective mean cell sizes, an estimation of the numerical uncertainty following Roache (1997) and execution times (excluding time for mesh generation, which may take several minutes) are presented in Tab. 1. Here, the runout is defined as the length of the central avalanche path (see Fig. 8) within the affected area. The central avalanche path has been taken from Fischer et al. (2015). The mean cell size is defined as the square root of the mean cell area. For comparison, execution times for SamosAT are 98 s (SPH-mode 0) and 368 s (SPH-mode 1), respectively. One should keep in mind that SamosAT utilises solely a single processor core while OpenFOAM utilises all available cores. Moreover, execution times should be seen as rough estimations because they depend on various factors, such as the number of saved time-steps, debug messages and compile options.

Deposition (i.e. flow thickness field $h$ in the last time-step) of the OpenFOAM solution is shown in Fig. 9 alongside with the documentation.

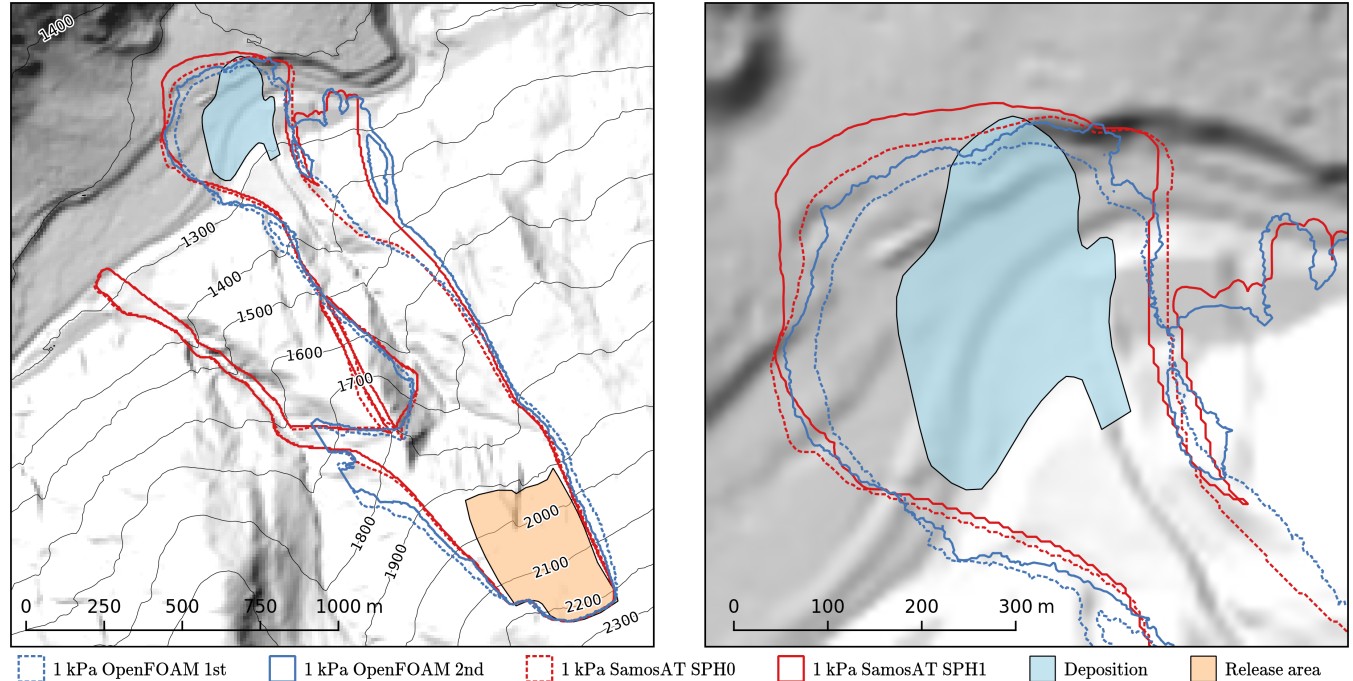

**Figure 7.** Comparison of OpenFOAM first-order (blue, dashed), OpenFOAM second-order (blue), SamosAT SPH 0 (red, dashed) and SamosAT SPH 1 (red) in terms of 1 kPa-isolines (affected area). OpenFOAM results are based on the mesh with cell size $\Delta = 7.45$ m. The documented release area (orange area) and documented deposition area (blue area) are shown for orientation. The shape and reach of the main avalanche branch are similar in all simulations, secondary branches differ to some extend. Overview (left) and focus on the runout zone (right). Terrain data: AdTLR.

## 4 Discussion and conclusion

Results of the new OpenFOAM solver are widely similar to SamosAT. Differences between SamosAT and OpenFOAM are in the range of numerical uncertainty and differences between interpolation methods are of comparable size. This uncertainty has to be expected; in fact, it is well known in the CFD community, that numerical schemes and implementation details influence results, if they are not converged to the analytical solution (e.g., Ferziger and Peric, 2002). In the case of gravitational mass flows, numerical uncertainty plays a minor role, since underlying models, parameters, terrain and snow cover data are afflicted with substantially higher uncertainty. This is shown by comparison of the documented deposition with the result of an OpenFOAM simulation in Fig. 9. Although parameters have been optimised to the specific event, all simulations differ significantly from documentation. Especially the large bulge on the orographic right side of the deposition area is not matched by any simulation. However, some details, such as the form of the tail and the position where the deposition expands, are accurately simulated by the OpenFOAM solver. Significant differences between simulation and documentation are not limited to the presented case and have been observed before by e.g. Rauter et al. (2016). We deduce that numerical errors are much

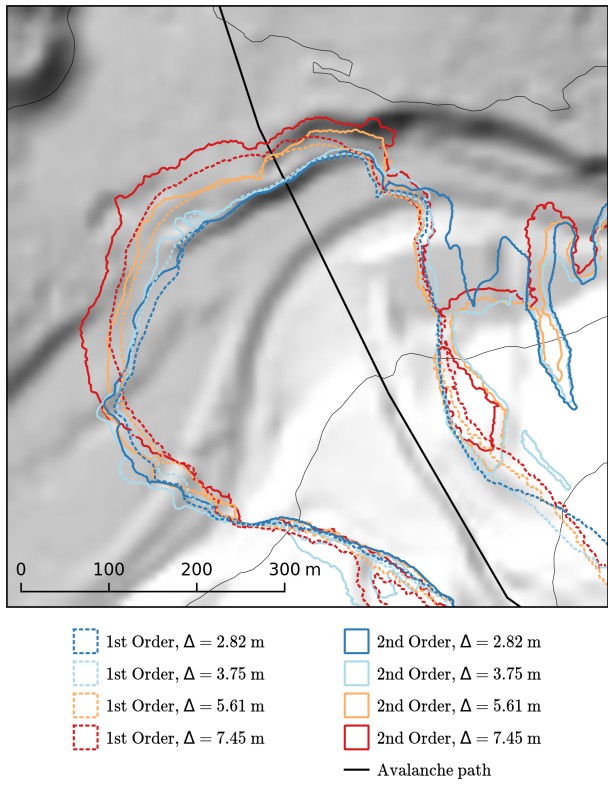

| | |
|---|---|
| ┄ 1st Order, $\Delta = 2.82$ m | ▢ 2nd Order, $\Delta = 2.82$ m |
| ┄ 1st Order, $\Delta = 3.75$ m | ▢ 2nd Order, $\Delta = 3.75$ m |
| ┄ 1st Order, $\Delta = 5.61$ m | ▢ 2nd Order, $\Delta = 5.61$ m |
| ┄ 1st Order, $\Delta = 7.45$ m | ▢ 2nd Order, $\Delta = 7.45$ m |
| | — Avalanche path |

**Figure 8.** Mesh refinement and convergence study for the OpenFOAM solver. Four mesh sizes and both interpolation schemes, first-order upwind (dashed line) and second-order Gamma (solid line) have been evaluated. The central avalanche path from Fischer et al. (2015) is shown in black. Terrain data: AdTLR.

smaller than the expected model error. Under these circumstances, a quantitative comparison between implementations (as by e.g. Rauter et al., 2016, for basal friction models) is not appropriate.

The refinement study shows that in the presented case, the simulated runout reduces with increasing mesh refinement (Fig. 8). Simulations on fine meshes are stopped by the first embankment, simulations on coarser grids overflow it and reach the next
5   embankment. This is reasonable, considering the higher diffusivity and lower curvature of coarser meshes. However, this trend should not be taken for granted for other cases and a refinement study should always be conducted to get an insight into the numerical uncertainty. Results indicate that a cell size of approximately $3.75\,\mathrm{m}$ is required in OpenFOAM to achieve convergence with respect to practical applications. The numerical uncertainty can not be calculated for the coarsest and finest mesh, since three simulations are required to conduct a Richardson extrapolation. It has to be noted that all simulations are
10   based on the same DEM with a grid size of $10\,\mathrm{m}$. The influence of terrain model quality (see, e.g., Bühler et al., 2011) on simulation results is not investigated.

The execution time of the OpenFOAM solver is acceptable for coarse meshes but increases with the square of the number of cells, because the time-step duration has to be reduced similarly to cell size. The OpenFOAM solver is noticeably slower

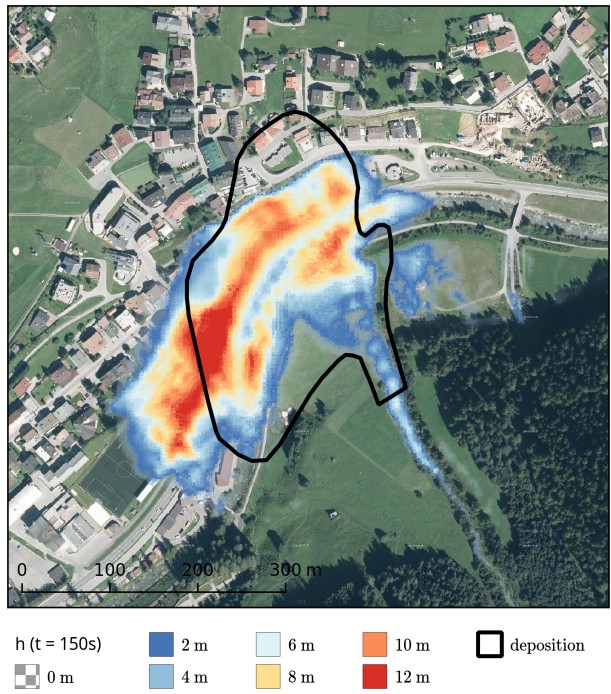

**Figure 9.** Flow thickness field $h$ at $t = 150\,\mathrm{s}$ of the second-order OpenFOAM simulation ($\Delta = 3.75\,\mathrm{m}$) and the documented deposition area. The flow thickness field in the last time step should roughly replicate the deposition. The bulge on the orographic right side of the deposition area is not matched by any simulation. However, some interesting details, such as the tail of the avalanche are represented well in OpenFOAM simulations. Map data: basemap.at.

than SamosAT, especially when considering OpenFOAMs multiprocessing capabilities. For applications where fast execution is imperative, such as parameter studies, SamosAT may be the appropriate choice. There is potential for future optimisation in OpenFOAM, especially the implicit time integration scheme is expensive and should be replaced with a simpler explicit one. However, the implicit solution strategy, in combination with the regularised friction relation, leads to a satisfying behaviour in the runout zone. In contrast, the simple explicit solution strategy from e.g. SamosAT leads to a continuous creeping of the deposition, meaning that the final flow thickness can not be compared with the deposition, as noticed by Fischer et al. (2015) and Rauter et al. (2016).

Stability of the OpenFOAM solver is strongly influenced by mesh quality. Simulations with polygonal-dominated surface meshes showed an acceptable stability for first- and second-order interpolations. The high influence of the three-dimensional mesh on stability and its computationally expensive creation is the main drawback of the proposed method. This is, however, also a big advantage, allowing simple coupling with three-dimensional ambient flows, as conducted by Sampl and Zwinger (2004).

## 5 Summary and Outlook

This paper shows the application of a finite area scheme for shallow granular flows (Rauter and Tuković, 2018) to snow avalanches on natural terrain. Specific processes, such as entrainment, have been added to the basic model to replicate the traditional model as implemented in SamosAT (Fischer et al., 2015).

Various simulations with the new OpenFOAM solver have been conducted. Methods and tools to incorporate the OpenFOAM solver in GIS have been presented. These tools allow integration of OpenFOAM in hazard mapping workflows and thus to validate the OpenFOAM solver with a reference implementation, herein SamosAT.

Application of three-dimensional Cartesian coordinates allows simple coupling with GIS applications because no coordinate transformations are required. Unstructured meshes, on the other hand, require re-sampling to structured meshes or data transfer in form of polygons. This incorporates an additional effort compared to simulations on structured meshes, as conducted by e.g. Christen et al. (2010).

The OpenFOAM solver roughly reproduces results of SamosAT. Differences are within the expected numerical uncertainty. A comparison of numerical results to a documented event suggests that model uncertainty is substantially higher than numerical uncertainties.

The major advantage of OpenFOAM is the object-oriented open-source code, which can be easily extended. The flexible code structure allows fast application of new models to real case examples. This especially qualifies the proposed method for model development and academic purposes. Moreover, the vast majority of source code is shared within the OpenFOAM community, leading to faster development of core features and higher code quality.

The finite area scheme allows a description in terms of surface partial differential equations (Deckelnick et al., 2005), which leads to simple and expressive governing equations. However, this comes at the cost of a complex three-dimensional surface mesh. Projection of the governing equations on a plane surface following e.g. Bouchut and Westdickenberg (2004) may be beneficial for some applications. The three-dimensional surface mesh can also be an advantage, allowing a simple coupling with three-dimensional ambient two-phase models for powder clouds (Sampl and Zwinger, 2004). The presented meshing method, creating a finite volume and the corresponding finite area mesh, is viable for such simulations as well.

Future steps will incorporate optimisation of the solver in terms of stability and execution time. Mesh generation and the integration of geographic information systems will be further streamlined. The limitation to mildly curved terrain should be eliminated, as this assumption is violated in many practical cases. We aim to implement more complex models, suitable for mixed snow avalanches (e.g., Bartelt et al., 2015; Issler et al., 2017) and debris flow (e.g., Iverson and George, 2014; Mergili et al., 2017) in the near future. Coupling of the here proposed dense flow model with three-dimensional two-phase models for the powder cloud regime (e.g. Cheng et al., 2017; Chauchat et al., 2017) is planned as well.

*Code and data availability.* The OpenFOAM solver, core utilities and the presented case study are available in the OpenFOAM community repository (https://develop.openfoam.com/Community/avalanche) and integrated as a module within OpenFOAM-v1712. The complete code

(based on foam-extend-4.0) including python scripts for GIS integration and the simulation setup including the underlying raw data is included in the supplementary material and available at https://bitbucket.org/matti2/fasavagehutterfoam.

## Appendix A: Understanding projections in surface partial differential equations

Here we shortly explain the concept of projections within the framework of surface partial differential equations. These projections are widely used in computational fluid dynamics, usually when surfaces in three dimensional space are considered. We do not focus on mathematical formalities and this section can not replace the formal derivation of Rauter and Tuković (2018). We want to emphasize that no surface aligned coordinate system is required throughout the whole process and the reader is encouraged to stick to global Cartesian coordinates. For simplicity we present a discretised finite area cell, which has been extruded by flow thickness $h$ to present the flowing mass, see Fig. A1.

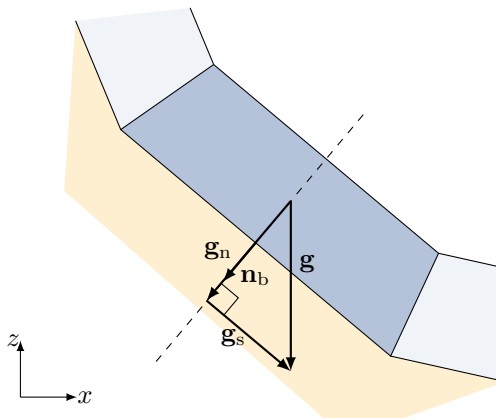

**Figure A1.** Splitting gravitational acceleration into a surface tangential and surface normal part with simple projections to the surface normal vector $\mathbf{n}_b$.

We begin by splitting a simple vectorial entity, the gravitational acceleration $\mathbf{g} \in \mathbb{R}^3$, into a surface normal component, $\mathbf{g}_n \in \mathbb{R}^3$, and a surface tangential component, $\mathbf{g}_s \in \mathbb{R}^3$, as shown in Fig. A1. The magnitude of the surface normal component can be calculated using the scalar-product and the surface normal vector,

$$\|\mathbf{g}_n\| = \mathbf{n}_b \cdot \mathbf{g}, \tag{A1}$$

which corresponds to a projection of $\mathbf{g}$ on $\mathbf{n}_b$. The surface normal component points in the same direction as the surface normal vector, which allows calculation of the vectorial surface normal component. Rearranging of vector multiplications yields the known form,

$$\mathbf{g}_n = \mathbf{n}_b \|\mathbf{g}_n\| = \mathbf{n}_b (\mathbf{n}_b \cdot \mathbf{g}) = (\mathbf{n}_b \, \mathbf{n}_b) \cdot \mathbf{g}. \tag{A2}$$

The surface tangential component follows by subtracting the surface normal component from total gravitational acceleration,

$$\mathbf{g}_s = \mathbf{g} - \mathbf{g}_n = \mathbf{g} - (\mathbf{n}_b \, \mathbf{n}_b) \cdot \mathbf{g} = (\mathbf{I} - \mathbf{n}_b \, \mathbf{n}_b) \cdot \mathbf{g}. \tag{A3}$$

Movement in surface normal direction is constrained by the basal topography, which yields the basal pressure. Therefore, the surface normal component $\mathbf{g}_n$ has to contribute to basal pressure $p_b$ (Eq. 3), and only the surface tangential component contributes to local acceleration $\frac{\partial h \overline{\mathbf{u}}}{\partial t}$ (Eq. 2). The total gravitational acceleration can be reconstructed by summing up both components,

$$\mathbf{g} = \mathbf{g}_n + \mathbf{g}_s = (\mathbf{n}_b \, \mathbf{n}_b) \cdot \mathbf{g} + (\mathbf{I} - \mathbf{n}_b \, \mathbf{n}_b) \cdot \mathbf{g} = \mathbf{I} \cdot \mathbf{g} = \mathbf{g}, \tag{A4}$$

reassuring perfect conservation of three dimensional momentum.

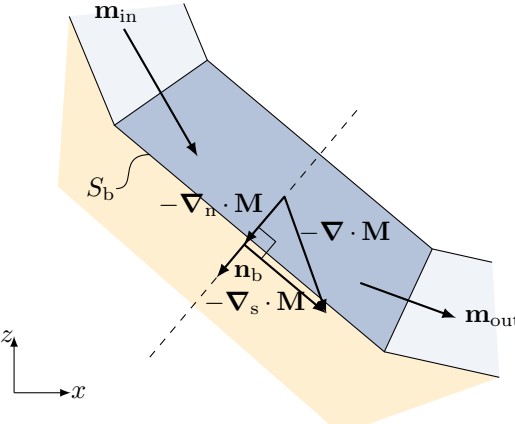

**Figure A2.** Splitting the divergence of a flux tensor $\boldsymbol{\nabla} \cdot \mathbf{M}$ into a surface tangential and surface normal part with simple projections to the surface normal vector $\mathbf{n}_b$.

The same concept can be applied to fluxes through the boundary of the control volume, leading to the concept of surface partial differential operators, $\boldsymbol{\nabla}_s$ and $\boldsymbol{\nabla}_n$. Figure A2 shows the divergence of a tensor, $\boldsymbol{\nabla} \cdot \mathbf{M}$, which could represent convective momentum transport $\boldsymbol{\nabla} \cdot (h \, \overline{\mathbf{u}} \, \overline{\mathbf{u}})$ or lateral pressure gradient $\frac{1}{2\rho} \boldsymbol{\nabla} (p_b \, h) = \frac{1}{2\rho} \boldsymbol{\nabla} \cdot (\mathbf{I} p_b \, h)$. Using Gauss' theorem, the divergence can be reformulated in terms of the surface integral of face fluxes, which are defined as the scalar product of the flux tensor $\mathbf{M}$ with the normal vector on the face (Ferziger and Peric, 2002). In the discretized form, integrals are replaced with sums over faces and in the case of SPDEs, volumes collapse to surfaces, faces to edges and face fluxes to edge fluxes. For the simple case, as shown in Fig. A2, we can write,

$$\boldsymbol{\nabla} \cdot \mathbf{M} = \frac{1}{S_b} \left( \mathbf{m}_{out} - \mathbf{m}_{in} \right), \tag{A5}$$

with area of the cell $S_b$ and edge fluxes $\mathbf{m}_{in}$ and $\mathbf{m}_{out}$. For the exact formulation in terms of finite areas, the reader is refereed to Rauter and Tuković (2018). Note that $\boldsymbol{\nabla} \cdot \mathbf{M}$ is a three dimensional vector without any particular direction in relation to the

basal surface. Hence, it has a surface tangential and a surface normal component which can be treated similar to gravitational acceleration, yielding the surface normal component

$$\boldsymbol{\nabla}_{\mathrm{n}} \cdot \mathbf{M} = \mathbf{n}_{\mathrm{b}} \, \|\boldsymbol{\nabla}_{\mathrm{n}} \cdot \mathbf{M}\| = \mathbf{n}_{\mathrm{b}} \, (\mathbf{n}_{\mathrm{b}} \cdot \boldsymbol{\nabla} \cdot \mathbf{M}) = (\mathbf{n}_{\mathrm{b}} \, \mathbf{n}_{\mathrm{b}}) \cdot \boldsymbol{\nabla} \cdot \mathbf{M}, \tag{A6}$$

and the surface tangential component

$$5 \quad \boldsymbol{\nabla}_{\mathrm{s}} \cdot \mathbf{M} = \boldsymbol{\nabla} \cdot \mathbf{M} - \boldsymbol{\nabla}_{\mathrm{n}} \cdot \mathbf{M} = \boldsymbol{\nabla} \cdot \mathbf{M} - (\mathbf{n}_{\mathrm{b}} \, \mathbf{n}_{\mathrm{b}}) \cdot \boldsymbol{\nabla} \cdot \mathbf{M} = (\mathbf{I} - \mathbf{n}_{\mathrm{b}} \, \mathbf{n}_{\mathrm{b}}) \cdot \boldsymbol{\nabla} \cdot \mathbf{M}. \tag{A7}$$

Surface normal and tangential components contribute to local acceleration and basal pressure for reasons discussed in terms of gravitational acceleration. Three dimensional conservation is reassured for fluxes as well, if $\boldsymbol{\nabla} \cdot \mathbf{M}$ is calculated conservatively. Finally, we want to note that velocity is a three-dimensional vector field and its direction is not fixed a priori. However, velocity will always be aligned with the surface because only surface tangential components are present in the respective conservation

10 equation.

*Competing interests.*  The authors declare that they have no conflict of interest.

*Acknowledgements.*  We thank Mark Olesen and Andrew Heather (ESI-OpenCFD) for help regarding OpenFOAM and review of our solver code. We thank Matthias Granig and Felix Oesterle (WLV) for support regarding SamosAT and for providing the respective software. We thank our colleges, Iman Bathaeian, Jan-Thomas Fischer and Fabian Schranz for valuable comments on the manuscript. We thank the

15 OpenFOAM, ParaView and QGIS communities for sharing their code and providing helpful advice. We further thank Stefan Hergarten, Julia Kowalski and one anonymous reviewer for their valuable comments which helped to increase clarity and quality of this paper. We gratefully acknowledge the financial support by the OEAW project "beyond dense flow avalanches". The computational results presented have been achieved (in part) using the HPC infrastructure LEO of the University of Innsbruck.

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
