# Peer review of "faSavageHutterFOAM 1.0: Depth-integrated simulation of dense snow avalanches on natural terrain with OpenFOAM"

_Geoscientific Model Development, 2018_

## Referee Comment (RC1) · Anonymous Referee #1 · 29 Mar 2018

This paper describes a numerical implementation of a snow avalanche model on complex natural topography within the OpenFoam open source toolkit. The topography is written in terms of surface partial differential equations, which makes extension to complex terrain straightforward within OpenFoam. This seems a nice way of programming the topographic effects. An example calculation is shown for the Wolfsgruben avalanche. The authors focus on an implementation of the Savage & Hutter (1989) model that has been generalized to 2D and contains many non-standard features, that the authors have programmed up, but I wonder how well they actually model the real physical system? There is a danger here that by providing an easily useable code it will receive widespread use by others and that the physical description gets engrained

in the community without adequate testing or questioning of the physics in the model.

1/. Equation 1 introduces a mass source through basal entrainment, but there seems to be no corresponding momentum source/sink in equation 2, is this correct? Surely there is some momentum exchange with the base, and it might be different depending on whether the avalanche is eroding or depositing material.

2/. Field studies of [SOVILLA, B., BURLANDO, P. & BARTELT, P. 2006 Field experiments and numerical modeling of mass entrainment in snow avalanches. J. Geophys. Res. 111, F03007; SOVILLA, B., SOMMAVILLA, F. & TOMASELLI, A. 2001 Measurements of mass balance in dense snow avalanche events. Ann. Glaciol. 32, 230–236] seem to suggest that most of the entrainment occurs by frontal ploughing rather than basal erosion. In fact some of the earliest snow avalanche models [BRIUKHANOV, A. V., GRIGORIAN, S. S., MIAGKOV, S. M., PLAM, M. Y., SHUROVA, I. E., EGLIT, M. E. & YAKIMOV, Y. L. 1967 On some new approaches to the dynamics of snow avalanches. In Physics of Snow and Ice, Proceedings of the International Conference on Low Temperature Science, vol. 1, pp. 1221–1241. Institute of Low Temperature Science, Hokkaido University] modelled the entrainment between a shallow water-like avalanche and a static layer of snow in front of it by using jump conditions across the interface to couple the two domains. Isn't this a better approach?

3/. I think the only way of really testing whether the entrainment model is any good is to compare it to carefully controlled laboratory experiments where the amount of entrainable material is known and there are key morphological features than can be seen in the subsequent deposits [EDWARDS, A. N., VIROULET, S., KOKELAAR, B. P. & GRAY, J. M. N. T. 2017 Formation of levees, troughs and elevated channels by avalanches on erodible slopes. J. Fluid Mech. 823, 278–315.] I strongly suspect that the current model will not be able to capture any of these features. It should be noted that very similar deposit features are often seen in snow avalanches at the Vallee de al Sionne, as well as at many other sites such as the Geschinen avalanche [ANCEY C. Are there "dragon-kings" events (i.e. genuine outliers) among extreme avalanches?

Eur. Phys. J. Special Topics 205, 117-129].

4/. One of the interesting features of erodible layers is that they appear to give avalanches far greater mobility (MANGENEY, A., ROCHE, O., HUNGR, O., MAN-GOLD, N., FACCANONI, G. & LUCAS, A. 2010 Erosion and mobility in granular collapse over sloping beds. J. Geophys. Res. 115, F03040), so there is a direct link to the apparent basal friction that the avalanche experiences. At the moment the drag law in equation (9) is completely decoupled from the erosion, so how can this be right?

5/. I also wonder how the basal friction law is able to keep material in the run-out zone stationary. As the velocity tends to zero, there is no basal friction, so I would expect the snow deposit to creep. The only way I can see things staying static is that everything has been deposited, i.e. the avalanche thickness is zero, can the authors clarify? This would actually speak in favour of having basal deposition as well as frontal ploughing.

6/. Coming back to the momentum balance (2) I note that the earth pressure coefficient has been neglected in the depth-integrated pressure gradient term. This was one of the key features that distinguished the Savage & Hutter (1989) theory from earlier Russian models. Is faSavageHutterFOAM 1.0: then a good name for the code? One of the problems of having the earth pressure coefficients is that they introduce stress discontinuities which generate shocks in the height and velocity of the flow. However, numerous observations [e.g. GRAY, J. M. N. T., TAI, Y. C. & NOELLE, S. 2003 Shock waves, dead-zones and particle-free regions in rapid granular free-surface flows. J. Fluid Mech. 491, 161–181; FAUG, T., CHILDS, P., WYBURN, E. & EINAV, I. 2015 Standing jumps in shallow granular flows down smooth inclines. Phys. Fluids 27, 073304.] demonstrate that only shocks that are expected from shallow-water/hydraulic type avalanche models are observed. Certainly there is no evidence for stress shocks from switching discontinuously from active and passive earth pressure regimes.

7/. There is no vertical acceleration in equation (3). Is there a good reason for that?

8/. In equation (4) the integrand is u(x_b-n_b,z'). I don't understand why we need the

-n_b term. Why not just u(x,b,z')?

9/.  At the bottom of the page the definition is made that $uv=u\otimes v$ where $\otimes$ is the dyadic product.  I think this contraction in notation is not good - I would recommend retaining the dyadic product $\otimes$ in the paper.  This is because at the moment matrix-vector multiplication is now shown with a dot product in equations (5-8). Just keep the standard notation.

10/.  Are equation (5)-(8) right?  I looked at the Deckelnick et al (2005) reference and found the equations the authors mention hard to find.  Just working out the tangential and normal components of the gravity vector g in a coordinate system aligned with the slope (such as that used by Savage & Hutter 1989) seems to give me some results that I find hard to interpret. It seems this gives a slope normal and tangential representation that is then translated back into a Cartesian coordinate system aligned with gravity. If I'm struggling here, then I suspect others will also.

11/.  Isn't a consequence of the friction law (9) that one gets steady states in which u∼sqrt(h)?  This seems to conflict with small experiments that show u∼hˆ(3/2) [POULIQUEN, O. & FORTERRE, Y. 2002 Friction law for dense granular flows:  application to the motion of a mass down a rough inclined plane. J. Fluid Mech. 453, 133–151]. Is there any evidence for this scaling?

12/. page 6, lines 10-15: "First-order schemes converge slower in terms of mesh refinement due to their high numerical diffusivity.  However, numerical diffusivity effectively prevents oscillations and increases the stability of the solver.  Oscillations in second-order accurate simulations are prevented with a normalised variable diagram (NVD) scheme for unstructured meshes, known as Gamma scheme (Jasak et al., 1999)." I wonder whether the oscillations that are being prevented are purely numerical? There can also be physical oscillations that are related to roll-waves which are observed in avalanches at a range of scale [FORTERRE, Y. & POULIQUEN, O. 2003 Long-surface-wave instability dense granular flows. J. Fluid Mech. 486, 21–50.; Kohler, A et al 2016

[Figure]

The dynamics of surges in the 3 February 2015 avalanches in Vallee de la Sionne JOURNAL OF GEOPHYSICAL RESEARCH-EARTH SURFACE Volume: 121 Issue: 11 Pages: 2192-2210; RAZIS et al. 2014 Arrested coarsening of granular roll waves. Phys. Fluids 26, 123305.] Are these being suppressed artificially by the numerics? Or does the friction law (9) not generate roll-waves?

13/. There are no scales in figures 3-6 or indication of the change in topographic height. The authors are probably so familiar with the simulations that this is obvious to them, but it is not obvious to the reader.

14/. page 12 line 11: modulus signs are missing on $|\bar u|<$ ...

15/. Figures 7-9: I know making comparisons to deposits from natural events is widespread in the literature, but can one really learn very much from this type of simulation and comparison. There seems so much physics that we don't understand properly and so much uncertainty about the initial conditions, snow entrainment and feedback on friction, that it is not surprising that we can't model the deposits accurately. Furthermore the friction coefficients can always be adjusted to get us in the right ball park for the run-out. Isn't there a more fundamental comparison that can be made?
* * *

---

## Referee Comment (RC2) · S. Hergarten (Referee) · 6 Apr 2018

The authors present an implementation of a model for rapid mass movements similar to the established Savage-Hutter model in the open-source continuum fluid dynamics software OpenFOAM. From the concept such an approach is somewhere between using highly developed specific models (which are mostly not free and not open source) and developing an own code from the scratch. My own 2015 paper already cited was a first approach in this direction, and the promising results of the recent manuscript illustrate that such concept could indeed become an interesting alternative in the future.

The biggest part of the implementation has already been published very recently in the

paper in Comp. Fluids. What is new here is the application to real topographies and the implementation of particle entrainment for snow avalanches. I feel that these new components indeed merit a publication in GMD.

The paper is well written in my opinion, and I enjoyed reading it. Unfortunately I did not find enough time to get as deep into the theory as the first reviewer did and can provide only a few suggestions where the presentation could be improved.

(i) As a principal problem, it was impossible to me to understand the brief review of the theory without the much longer paper in Comp. Fluids. However, I do not know whether there is a way to get around this problem, but maybe the authors can think about it.

(ii) In the beginning I got stuck at the way how the fundamental equations (Eqs. 1-3) have been extended by particle entrainment, in particular why only the first equation is affected. I think it is correct this way using the momentum-flux version of the shallow water equations, but maybe a bit more explanation on this might help.

(iii) How does the "non-physical" parameter $u\_0$ affect the results, in particular the question how close the avalanche comes to rest?

(iv) As a detail of the implementation, I did not get how the regularisation of Eq. 12 similar to Eq. 9 works.

I hope that these suggestions help in improving the accessibility of the paper for those readers who are not so familiar with the theory.

---

## Author Comment (AC1) · 25 Apr 2018

This paper describes a numerical implementation of a snow avalanche model on complex natural topography within the OpenFoam open source toolkit. The topography is written in terms of surface partial differential equations, which makes extension to complex terrain straightforward within OpenFoam. This seems a nice way of programming the topographic effects. An example calculation is shown for the Wolfsgruben avalanche. The authors focus on an implementation of the Savage & Hutter (1989) model that has been generalized to 2D and contains many non-standard features, that the authors have programmed up, but I wonder how well they actually model the real

physical system? There is a danger here that by providing an easily useable code it will receive widespread use by others and that the physical description gets engrained in the community without adequate testing or questioning of the physics in the model.

Dear Referee,

thank you very much for your fast review and your interesting remarks on the manuscript. We share many of your concerns regarding the accuracy of the physical model and the validity of friction and entrainment relations. However, the applied flow and process models present a setup that is currently widely used, as shown in a recent survey [Schmidtner, Korbinian; Sailer, Rudolf; Bartelt, Perry; Fellin, Wolfgang; Fischer, Jan-Thomas; Granig, Matthias (2017): Investigations on the Application of Avalanche Simulations: A Survey Conducted among Avalanche Experts. In: ICNDAHR 2017: International Conference on Natural Disasters, Assessing Hazards and Risk, London, United Kingdom, (May 25-26, 2017). Canakkale: World Academy of Science, Engineering and Technology (WASET)]. We stick to the "standard setup" to not get lost in the discussion about appropriate closure relations and to focus on the general framework of faSavageHutterFOAM. Note that various alternative friction and entrainment models can be selected by the user (See page 5, lines 17ff). This includes friction models fulfilling Bagnold scaling ($\overline{u} \sim h^{(3/2)}$) and a front entrainment model, as you propose in your comments. We refer to the manual within supplementary materials for details. Moreover, the open structure allows simple application of custom models. This is, in our opinion, especially important, allowing everybody to apply the preferred friction and entrainment models. Finally, we want to note that developing and evaluating more realistic process models requires a platform for testing, which the proposed solver is intended to provide.

1/. Equation 1 introduces a mass source through basal entrainment, but there seems to be no corresponding momentum source/sink in equation 2, is this correct? Surely

there is some momentum exchange with the base, and it might be different depending on whether the avalanche is eroding or depositing material.

Entrained mass has no velocity at the time of erosion. The respective momentum is therefore zero and no momentum source has to be taken into account. Note that we use the conservative description. In a non-conservative formulation, an entrainment term will show up in the momentum equation. Momentum exchange with the base is considered with the basal friction term $\tau_b$. The interaction between entrainment and rheology is usually not considered for practical applications, although considered important by many scientists. We are aware of such approaches and they can be easily implemented in faSavageHutterFOAM.

2/. Field studies of [SOVILLA, B., BURLANDO, P. & BARTELT, P. 2006 Field experiments and numerical modeling of mass entrainment in snow avalanches. J. Geophys. Res. 111, F03007; SOVILLA, B., SOMMAVILLA, F. & TOMASELLI, A. 2001 Measurements of mass balance in dense snow avalanche events. Ann. Glaciol. 32, 230–236] seem to suggest that most of the entrainment occurs by frontal ploughing rather than basal erosion. In fact some of the earliest snow avalanche models [BRIUKHANOV,A. V., GRIGORIAN, S. S., MIAGKOV, S. M., PLAM, M. Y., SHUROVA, I. E., EGLIT, M. E. & YAKIMOV, Y. L. 1967 On some new approaches to the dynamics of snow avalanches. In Physics of Snow and Ice, Proceedings of the International Conference on Low Temperature Science, vol. 1, pp. 1221–1241. Institute of Low Temperature Science, Hokkaido University] modelled the entrainment between a shallow water-like avalanche and a static layer of snow in front of it by using jump conditions across the interface to couple the two domains. Isn't this a better approach?

Such models can be implemented. See answer below.

3/. I think the only way of really testing whether the entrainment model is any good is to compare it to carefully controlled laboratory experiments where the amount of entrainable material is known and there are key morphological features than can be seen in the subsequent deposits [EDWARDS, A. N., VIROULET, S., KOKELAAR, B. P. & GRAY, J. M. N. T. 2017 Formation of levees, troughs and elevated channels by avalanches on erodible slopes. J. Fluid Mech. 823, 278–315.] I strongly suspect that the current model will not be able to capture any of these features. It should be noted that very similar deposit features are often seen in snow avalanches at the Vallee de al Sionne, as well as at many other sites such as the Geschinen avalanche [ANCEY C. Are there "dragon-kings" events (i.e. genuine outliers) among extreme avalanches? Eur. Phys. J. Special Topics 205, 117-129].

We agree that the shown model may be to simple to model such effects. More elaborated models can be implemented in the code (some are already available). However, it is out of the scope of this paper to compare different models. This will be undertaken in future research.

4/. One of the interesting features of erodible layers is that they appear to give avalanches far greater mobility (MANGENEY, A., ROCHE, O., HUNGR, O., MANGOLD, N., FACCANONI, G. & LUCAS, A. 2010 Erosion and mobility in granular collapse over sloping beds. J. Geophys. Res. 115, F03040), so there is a direct link to the apparent basal friction that the avalanche experiences. At the moment the drag law in equation (9) is completely decoupled from the erosion, so how can this be right?

The evaluation of most friction and entrainment models is still missing. We stick in this paper to the simplest and commonly applied relations as the main goal was to show the basic performance of the new implementation method in comparison to well established ones. Various entrainment models (including front entrainment) are

available in the presented code and can be selected in transportProperties. Moreover, the framework is designed to simplify the implementation of new entrainment and friction models. See manual within supplementary materials for details. This will hopefully help to select the appropriate physical models in the near future.

5/. I also wonder how the basal friction law is able to keep material in the run-out zone stationary. As the velocity tends to zero, there is no basal friction, so I would expect the snow deposit to creep. The only way I can see things staying static is that everything has been deposited, i.e. the avalanche thickness is zero, can the authors clarify? This would actually speak in favour of having basal deposition as well as frontal ploughing.

You are correct. However, the creeping velocity is approximately $10^{-5}$m/s. This value is lower than the tolerance of the solver and thus virtually zero, especially when compared to the creeping velocity of other implementations. We will add the respective remark to the revised manuscript.

6/. Coming back to the momentum balance (2) I note that the earth pressure coefficient has been neglected in the depth-integrated pressure gradient term. This was one of the key features that distinguished the Savage & Hutter (1989) theory from earlier Russian models. Is faSavageHutterFOAM 1.0: then a good name for the code? One of the problems of having the earth pressure coefficients is that they introduce stress discontinuities which generate shocks in the height and velocity of the flow. However, numerous observations [e.g. GRAY, J. M. N. T., TAI, Y. C. & NOELLE, S. 2003 Shock waves, dead-zones and particle-free regions in rapid granular free-surface flows. J. Fluid Mech. 491, 161–181; FAUG, T., CHILDS, P., WYBURN, E. & EINAV, I. 2015 Standing jumps in shallow granular flows down smooth inclines. Phys. Fluids 27, 073304.] demonstrate that only shocks that are expected from shallow-water/hydraulic type avalanche models are observed. Certainly there is no evidence for stress shocks

from switching discontinuously from active and passive earth pressure regimes.

We are aware of the fact that contributions to depth-integrated avalanche modelling are highly controversial. We try to accurately describe the historical developments in the introduction. We try to attribute accomplishments to the appropriate groups to the best of our ability and the Russian models are mentioned as well.

We think that the work of Savage and Hutter (1989) has been one of the most influential ones, introducing depth-integrated avalanche simulations to a wide audience. The name Savage-Hutter model has become a synonym for depth-integrated avalanche models, which is also acknowledged by competing scientists, e.g. [Salm, Bruno. "A short and personal history of snow avalanche dynamics." Cold Regions Science and Technology 39.2-3 (2004): 83-92.]: *"Here I would like to pause and emphasize that as early as 1966, the foundation of what is now known as the Savage–Hutter theory (Savage and Hutter, 1989) was squarely in place."*

The decisive difference between the Savage-Hutter-model and other depth-integrated models in OpenFOAM is the assumed direction of the velocity (horizontal vs. surface tangetial). We use the respective name to distinguish our model from strictly two-dimensional ones with horizontal velocity (e.g. ShallowFOAM, https://github.com/mintgen/shallowFoam).

7/. There is no vertical acceleration in equation (3). Is there a good reason for that?

The surface normal acceleration of the flow is zero, according to the model assumption of surface tangential velocity, $(\mathbf{n}_b\,\mathbf{n}_b)\cdot\overline{\mathbf{u}} = \mathbf{0}$, see Rauter and Tukovic (2018). Thus no acceleration is present in Eq. (3). The vertical acceleration is included in Eq. (2) as part of the surface tangential acceleration, $\frac{\partial}{\partial t}\left(h\,\overline{\mathbf{u}}\right) = \frac{\partial}{\partial t}\left(h\left(\overline{u}_x, \overline{u}_y, \overline{u}_z\right)^{\mathsf{T}}\right)$.
8/. In equation (4) the integrand is $u(x_b - n_b z')$. I don't understand why we need the $-n_b$ term. Why not just $u(x, b, z')$?

The function $\mathbf{u}(\mathbf{x})$ maps global Cartesian coordinates $\mathbf{x}$ to the velocity in the same coordinate system. The notation including the surface normal vector $\mathbf{n}_b$ was proposed by a referee of Rauter and Tukovic (2018). The surface normal vector should indicate the direction of depth-integration, which is surface normal.

9/. At the bottom of the page the definition is made that $uv = u \otimes v$ where $\otimes$ is the dyadic product. I think this contraction in notation is not good - I would recommend retaining the dyadic product $\otimes$ in the paper. This is because at the moment matrix-vector multiplication is now shown with a dot product in equations (5-8). Just keep the standard notation.

The current notation has been adopted from common literature on the subject (e.g. Craster and Matar, 2009; Tukovic and Jasak, 2012; Rauter and Tukovic 2018) and we would like to be consistent with them.

10/. Are equation (5)-(8) right? I looked at the Deckelnick et al (2005) reference and found the equations the authors mention hard to find. Just working out the tangential and normal components of the gravity vector g in a coordinate system aligned with the slope (such as that used by Savage & Hutter 1989) seems to give me some results that I find hard to interpret. It seems this gives a slope normal and tangential representation that is then translated back into a Cartesian coordinate system aligned with gravity. If I'm struggling here, then I suspect others will also.

Equations (5) and (6) are simple projections, which are presented for a didac-

tic reason. They introduce the projection of vectorial entities on a surface. This is a common procedure in CFD [e.g., Vukcevic, V. Numerical modelling of coupled potential and viscous flow for marine applications. PhD thesis, Faculty of Mechanical Engineering and Naval Architecture, University of Zagreb, 2016., https://dx.doi.org/10.13140/RG.2.2.23080.57605, page 16] and showing the example with the gravitational acceleration makes the concept easier to grasp. Equation (7) can be found in Deckelnick et al (2005), section 2.1 (note that variable names are different) and Eq. (8) is an extension, which has been introduced by Rauter and Tukovic (2018).

No surface aligned coordinate system is required to work out the projects. In fact, thinking in surface aligned coordinates is not suitable to understand the presented method.

We propose to add a simple introduction to the applied projections to the appendix of the manuscript (see appended document).

11/. Isn't a consequence of the friction law (9) that one gets steady states in which $u \sim sqrt(h)$? This seems to conflict with small experiments that show $u \sim h^{(3/2)}$ [POULIQUEN, O. & FORTERRE, Y. 2002 Friction law for dense granular flows: application to the motion of a mass down a rough inclined plane. J. Fluid Mech. 453, 133–151]. Is there any evidence for this scaling?

The applied friction law does not obey Bagnold scaling ($\overline{u} \sim h^{(3/2)}$), that is correct. We are aware of this issue and used other friction laws, which obey Bagnold scaling, in previous works (see Rauter et al., 2016; Rauter and Tukovic, 2018). As mentioned in the manuscript (page 5, line 17ff), we chose the Voellmy friction relation to resemble the traditional model. Various friction laws are available in the code and can be selected by the user (see manual within supplementary materials for details). This includes various friction laws which obey Bagnold scaling.

12/. page 6, lines 10-15: "First-order schemes converge slower in terms of mesh refinement due to their high numerical diffusivity. However, numerical diffusivity effectively prevents oscillations and increases the stability of the solver. Oscillations in second-order accurate simulations are prevented with a normalised variable diagram (NVD) scheme for unstructured meshes, known as Gamma scheme (Jasak et al., 1999)." I wonder whether the oscillations that are being prevented are purely numerical? There can also be physical oscillations that are related to roll-waves which are observed in avalanches at a range of scale [FORTERRE, Y. & POULIQUEN, O. 2003 Long-surface-wave instability dense granular flows. J. Fluid Mech. 486, 21–50.; Kohler, A et al 2016 The dynamics of surges in the 3 February 2015 avalanches in Vallee de la Sionne JOURNAL OF GEOPHYSICAL RESEARCH-EARTH SURFACE Volume: 121 Issue: 11 Pages: 2192-2210; RAZIS et al. 2014 Arrested coarsening of granular roll waves. Phys. Fluids 26, 123305.] Are these being suppressed artificially by the numerics? Or does the friction law (9) not generate roll-waves?

The application of the discussed schemes is imperative for the solution of hyperbolic partial differential equations and generally accepted in CFD. Schemes are designed to solely suppress numerical oscillations, although this is difficult and not always possible. The applied friction law does not generate roll-waves. Therefore, oscillations have to be purely numerical (see, e.g. Wang et al., 2004, for a demonstration). The solver should be able to generate roll-waves with other friction laws.

13/. There are no scales in figures 3-6 or indication of the change in topographic height. The authors are probably so familiar with the simulations that this is obvious to them, but it is not obvious to the reader.

That is a valid point. However, in Figure 3 (x,y,z)-coordinates in the co-ordinate reference system "MGI/ Austria GK West" (EPSG: 31254) (see: http://spatialreference.org/ref/epsg/31254/) are provided for each subfigure - the z coordinate gives an indication for the change in topographic height for the calculation domain from release area to the valley bottom. Scales for horizontal distances are provided in Figures 3,6,7,8 and 9. In the revised version we will add labels to the contour lines in figure 7 for clarification of topographic height changes.

14/. page 12 line 11: modulus signs are missing on $|\bar{u}| < ...$

This is correct. We will add modulus signs in the revised manuscript.

15/. Figures 7-9: I know making comparisons to deposits from natural events is widespread in the literature, but can one really learn very much from this type of simulation and comparison. There seems so much physics that we don't understand properly and so much uncertainty about the initial conditions, snow entrainment and feedback on friction, that it is not surprising that we can't model the deposits accurately. Furthermore the friction coefficients can always be adjusted to get us in the right ball park for the run-out. Isn't there a more fundamental comparison that can be made?

We are aware of the manifold ways to compare simulations. However, the main intention of the paper is to present the new solver and to show that existing results (of e.g. SamosAT) can be roughly reproduced.

Comparisons to analytical solutions can be found in Rauter and Tukovic (2018).

Moveover, we conducted simulations of avalanches from test sites Ryggfonn and Valleé de la Sionne, including comparisons to radar measurments [Rauter, M. (2017):

A finite area scheme for shall ow granular flows on three-dimensional surfaces. Poster presented at: European Geosciences Union (EGU) General Assembly 2017, Wien, 24.04.2017; Rauter, M., Köhler, A. (2017): A Finite Area Scheme for Shallow Granular Flows. Presented at: 12th Workshop on OpenFOAM (OFW12), Exeter, 26.07.2017]. However, methods for comparison are complex and would distract from the primary goal of the paper. Moreover, the underlying data is not freely available and therefore, not siuted for GMD, which highly values transparency.

---

## Author Comment (AC2) · 25 Apr 2018

The authors present an implementation of a model for rapid mass movements similar to the established Savage-Hutter model in the open-source continuum fluid dynamics software OpenFOAM. From the concept such an approach is somewhere between using highly developed specific models (which are mostly not free and not open source) and developing an own code from the scratch. My own 2015 paper already cited was a first approach in this direction, and the promising results of the recent manuscript illustrate that such concept could indeed become an interesting alternative in the future. The biggest part of the implementation has already been published very

recently in the paper in Comp. Fluids. What is new here is the application to real topographies and the implementation of particle entrainment for snow avalanches. I feel that these new components indeed merit a publication in GMD. The paper is well written in my opinion, and I enjoyed reading it. Unfortunately I did not find enough time to get as deep into the theory as the first reviewer did and can provide only a few suggestions where the presentation could be improved.

Dear Prof. Hergarten,

thank you very much for your quick review and your interesting suggestions. This paper is indeed a summary of open questions from referees of the last paper. Especially (1) application to natural terrain, (2) comparison to an existing software and (3) interaction with GIS was requested by the referees of Rauter and Tukovic (2018) for future publications.

(i) As a principal problem, it was impossible to me to understand the brief review of the theory without the much longer paper in Comp. Fluids. However, I do not know whether there is a way to get around this problem, but maybe the authors can think about it.

(ii) In the beginning I got stuck at the way how the fundamental equations (Eqs. 1-3) have been extended by particle entrainment, in particular why only the first equation is affected. I think it is correct this way using the momentum-flux version of the shallow water equations, but maybe a bit more explanation on this might help.

We are aware of the fact that the theory is hard to understand for readers which are used to the classic surface aligned coordinate system. We think the biggest problem is that people think in surface aligned coordinates and it usually takes a while to get used to the new concept of projections and the usage of Cartesian coordinates. To solve

this issue, we propose to add an explanation of surface projections in the appendix (see appended document).

(iii) How does the "non-physical" parameter $u_0$ affect the results, in particular the question how close the avalanche comes to rest?

The parameter $u_0$ has no relevant effect, as long as $|\overline{u}| \gg u_0$. At $|\overline{u}| = 100\,u_0$, the regularisation reduces the effective friction to $99\%$, at $|\overline{u}| = 10\,u_0$ to $91\%$, etc. This allows stopping in zones where only part of the possible shear stress is mobilized. We are using $u_0 = 10^{-7}$m/s in the simulations. Therefore, there shouldn't be any relevant effect on the dynamic behaviour at $|\overline{u}| > 10^{-5}$m/s. Indeed, $10^{-5}$m/s is the velocity we observe in the deposition zone. This velocity matches the solver tolerance, and can thus be ignored. We will add this to the revised manuscript.

(iv) As a detail of the implementation, I did not get how the regularisation of Eq. 12 similar to Eq. 9 works. I hope that these suggestions help in improving the accessibility of the paper for those readers who are not so familiar with the theory.

The regularisation works by reformulating the explicitly calculated entrainment rate to an implicit entrainment rate, depending on $h_{\mathrm{msc}}$,

$$\frac{\dot{q}}{\rho} = \frac{\dot{q}}{\rho}\,\frac{h_{\mathrm{msc}}}{h_{\mathrm{msc}} + h_0}.$$

The term $\frac{\dot{q}}{\rho}\,\frac{1}{h_{\mathrm{msc}}+h_0}$ is calculated (similar to $\mu\,p_{\mathrm{b}}\,\frac{1}{|\overline{\mathbf{u}}|+u_0}$) following the entrainment model, a small value for $h_0$ and an estimation for $h_{\mathrm{msc}}$. This way, the entrainment rate is reduced for $h_{\mathrm{msc}} \to 0$ and undershoots are prevented. Note that this works only when solving the equation implicitly. We will add this to the revised manuscript.

Please also note the supplement to this comment:
https://www.geosci-model-dev-discuss.net/gmd-2018-67/gmd-2018-67-AC2-supplement.pdf

**Supplement:**

The finite area scheme allows a description in terms of surface partial differential equations (Deckelnick et al., 2005), which leads to simple and expressive governing equations. However, this comes at the cost of a complex three-dimensional surface mesh. Projection of the governing equations on a plane surface following e.g. Bouchut and Westdickenberg (2004) may be beneficial for some applications. The three-dimensional surface mesh can also be an advantage, allowing a simple coupling with three-dimensional ambient two-phase models for powder clouds (Sampl and Zwinger, 2004). The presented meshing method, creating a finite volume and the corresponding finite area mesh, is viable for such simulations as well.

Future steps will incorporate optimisation of the solver in terms of stability and execution time. Mesh generation and the integration of geographic information systems will be further streamlined. We aim to implement more complex models, suitable for mixed snow avalanches (e.g., Bartelt et al., 2015; Issler et al., 2017) and debris flow (e.g., Iverson and George, 2014; Mergili et al., 2017) in the near future. Coupling of the here proposed dense flow model with three-dimensional two-phase models for the powder cloud regime (e.g. Cheng et al., 2017; Chauchat et al., 2017) is planned as well.

*Code and data availability.* The OpenFOAM solver, core utilities and the presented case study are available in the OpenFOAM community repository (https://develop.openfoam.com/Community/avalanche) and integrated as a module within OpenFOAM-v1712. The complete code (based on foam-extend-4.0) including python scripts for GIS integration and the simulation setup including the underlying raw data is included in the supplementary material and available at https://bitbucket.org/matti2/fasavagehutterfoam.

**Appendix A:  Understanding projections in surface partial differential equations**

Here we shortly explain the concept of projections within the framework of surface partial differential equations. These projections are widely used in computational fluid dynamics, usually when surfaces in three dimensional space are considered. We do not focus on mathematical formalities and this section can not replace the formal derivation of Rauter and Tuković (2018). We want to emphasize that no surface aligned coordinate system is required throughout the whole process and the reader is encouraged to stick to global Cartesian coordinates. For simplicity we present a discretised finite area cell, which has been extruded by flow thickness $h$ to present the flowing mass, see Fig. A1.

We begin by splitting a simple vectorial entity, the gravitational acceleration $\mathbf{g} \in \mathbb{R}^3$, into a surface normal component, $\mathbf{g}_\mathrm{n} \in \mathbb{R}^3$, and a surface tangential component, $\mathbf{g}_\mathrm{s} \in \mathbb{R}^3$, as shown in Fig. A1. The magnitude of the surface normal component can be calculated using the scalar-product and the surface normal vector,

$$\|\mathbf{g}_\mathrm{n}\| = \mathbf{n}_\mathrm{b} \cdot \mathbf{g}, \tag{A1}$$

which corresponds to a projection of $\mathbf{g}$ on $\mathbf{n}_\mathrm{b}$. The surface normal component points in the same direction as the surface normal vector, which allows calculation of the vectorial surface normal component. Rearranging of vector multiplications yields the known form,

$$\mathbf{g}_\mathrm{n} = \mathbf{n}_\mathrm{b} \|\mathbf{g}_\mathrm{n}\| = \mathbf{n}_\mathrm{b} \left( \mathbf{n}_\mathrm{b} \cdot \mathbf{g} \right) = \left( \mathbf{n}_\mathrm{b} \, \mathbf{n}_\mathrm{b} \right) \cdot \mathbf{g}. \tag{A2}$$

[Figure]

**Figure A1.** Splitting gravitational acceleration into a surface tangential and surface normal part with simple projections to the surface normal vector $\mathbf{n}_b$.

The surface tangential component follows by subtracting the surface normal component from total gravitational acceleration,

$$\mathbf{g}_s = \mathbf{g} - \mathbf{g}_n = \mathbf{g} - (\mathbf{n}_b \, \mathbf{n}_b) \cdot \mathbf{g} = (\mathbf{I} - \mathbf{n}_b \, \mathbf{n}_b) \cdot \mathbf{g}. \tag{A3}$$

Movement in surface normal direction is constrained by the basal topography, which yields the basal pressure. Therefore, the surface normal component $\mathbf{g}_n$ has to contribute to basal pressure $p_b$ (Eq. 3), and only the surface tangential component contributes to local acceleration $\frac{\partial h \overline{\mathbf{u}}}{\partial t}$ (Eq. 2). The total gravitational acceleration can be reconstructed by summing up both components,

$$\mathbf{g} = \mathbf{g}_n + \mathbf{g}_s = (\mathbf{n}_b \, \mathbf{n}_b) \cdot \mathbf{g} + (\mathbf{I} - \mathbf{n}_b \, \mathbf{n}_b) \cdot \mathbf{g} = \mathbf{I} \cdot \mathbf{g} = \mathbf{g}, \tag{A4}$$

reassuring perfect conservation of three dimensional momentum.

The same concept can be applied to fluxes through the boundary of the control volume, leading to the concept of surface partial differential operators, $\boldsymbol{\nabla}_s$ and $\boldsymbol{\nabla}_n$. Figure A2 shows a momentum flux $\boldsymbol{\nabla} \cdot \mathbf{m}$, which could represent convective momentum transport $\boldsymbol{\nabla} \cdot (h \, \overline{\mathbf{u}} \, \overline{\mathbf{u}})$ or lateral pressure gradient $\frac{1}{2\rho} \boldsymbol{\nabla} \, (p_b \, h)$. The net flux leaving the control volume can be calculated as the sum of all fluxes and leads to the definition of the divergence operator,

$$\boldsymbol{\nabla} \cdot \mathbf{m} = \frac{1}{S_b} \, (\mathbf{m}_{out} - \mathbf{m}_{in}), \tag{A5}$$

in accordance to Gauss' theorem. $S_b$ is the surface area of the cell. For the exact formulation in terms of finite areas, the reader is refereed to Rauter and Tuković (2018). Note that the net flux is a three dimensional vector without any particular direction in relation to the basal surface. Hence, it has a surface tangential component and a surface normal component. It can thus be treated similar to gravitational acceleration, yielding the surface normal component

$$\boldsymbol{\nabla}_n \cdot \mathbf{m} = \mathbf{n}_b \, \|\boldsymbol{\nabla}_n \cdot \mathbf{m}\| = \mathbf{n}_b \, (\mathbf{n}_b \cdot \boldsymbol{\nabla} \cdot \mathbf{m}) = (\mathbf{n}_b \, \mathbf{n}_b) \cdot \boldsymbol{\nabla} \cdot \mathbf{m}. \tag{A6}$$

[Figure]

**Figure A2.** Splitting net fluxes into a surface tangential and surface normal part with simple projections to the surface normal vector $\mathbf{n}_b$. Note that the flux, entering the control volume, $-\boldsymbol{\nabla}\cdot\mathbf{m}$, is shown.

and the surface tangential component

$$\boldsymbol{\nabla}_s\cdot\mathbf{m} = \boldsymbol{\nabla}\cdot\mathbf{m} - \boldsymbol{\nabla}_n\cdot\mathbf{m} = \boldsymbol{\nabla}\cdot\mathbf{m} - (\mathbf{n}_b\,\mathbf{n}_b)\cdot\boldsymbol{\nabla}\cdot\mathbf{m} = (\mathbf{I} - \mathbf{n}_b\,\mathbf{n}_b)\cdot\boldsymbol{\nabla}\cdot\mathbf{m}. \tag{A7}$$

Surface normal and tangential components contribute to local acceleration and basal pressure for reasons discussed in terms of gravitational acceleration. Three dimensional conservation is reassured for fluxes as well, if the three dimensional flux
5   $\boldsymbol{\nabla}\cdot\mathbf{m}$ is conservative. Finally, we want to note that velocity is a three-dimensional vector field and its direction is not fixed a priori. However, velocity will always be aligned with the surface because only surface tangential components are present in the respective conservation equation.

*Competing interests.*   The authors declare that they have no conflict of interest.

*Acknowledgements.*   We thank Mark Olesen and Andrew Heather (ESI-OpenCFD) for help regarding OpenFOAM and review of our solver
10   code. We thank Matthias Granig and Felix Oesterle (WLV) for support regarding SamosAT and for providing the respective software. We thank our colleges, Iman Bathaeian, Jan-Thomas Fischer and Fabian Schranz for valuable comments on the manuscript. We thank the OpenFOAM, ParaView and QGIS communities for sharing their code and providing helpful advice. We gratefully acknowledge the financial support by the OEAW project "beyond dense flow avalanches". The computational results presented have been achieved (in part) using the HPC infrastructure LEO of the University of Innsbruck.

---

## Referee Comment (RC3) · JK Kowalski (Referee) · 15 May 2018

**Review: faSavageHutterFOAM 1.0: Depth-integrated simulation of dense snow avalanches on natural terrain with OpenFOAM**

By Julia Kowalski

The manuscript *faSavageHutterFOAM 1.0: Depth-integrated simulation of dense snow avalanches on natural terrain with OpenFOAM* is a well-written contribution to the field of modeling and simulation of dense snow avalanches and fits into the scope of the journal. It extends the description of the OpenFOAM implementation first published by one of the authors, Matthias Rauter in (Rauter and Tukovic, 2018), to realistic topographies. The authors introduce the OpenFOAM specific workflow with a special focus on mesh generation from DEM data and GIS compatible post-processing. They demonstrate the capability of the proposed solver while analyzing a specific case study, namely the Wolfsgruben avalanche. Simulation results of the new OpenFOAM based solver are compared to the established tool samosAT.

My overall impression of this article is very positive due to the following reasons:

- First of all, by implementing the mathematical model into the mature software framework OpenFOAM, the authors actually outsource a lot of issues associated with e.g. data structure, parallelization and linear solvers. This allows a focus on the mathematical/physical model formulation itself, which is in my opinion the correct way to go.

- The mathematical/physical model itself is passed over to the solver in an encapsulated way as demonstrated on page 6 line 20 - 30. I find this a very big advantage of the approach. In fact this seems to be so straight forward, that one of the other reviewers even objected on the relative ease with which the model can be applied to real hazard scenarios, and the responsibility that arises from that. I certainly do have a slightly different opinion: While we as modelers do indeed have the responsibility to investigate and clearly communicate any limitations to the model, I fully support the authors approach in aiming towards modularizing software development and mathematical model representation as far as possible.

Thank you very much. Indeed, this was the main motivation of this work.

- Another aspect of the paper that I like very much is the comparison to an alternative simulation model. Although I have to admit that personally I find SamosAT a somewhat surprising choice (see my comment further down), this in principle is exactly the kind of results that help to better assess scope, potential and limitation of the various software tools that are in use.

- Finally, I want to emphasize the author's attempt to publish reproducible results. The paper refers to the code, which can be downloaded and tested.

All in all, I am clearly in favor of publishing this paper and think it will be a valuable contribution to the field, if the following objections have been addressed.

- Title seems odd: Probably the title is chosen according to the OpenFOAM module - still I think it doesn't reflect the content of the paper. While the mechanical model that the authors solve is similar to the Savage-Hutter model (see also my next comment), some of the original Savage Hutter theory's defining flavors, e.g. its questionable active-passive

earth-pressure term and the 'dry-friction-only' resistive force, are absent. This , however, is also OK, as it does not seem to be what the authors want to promote, compare and discuss in their paper. Straight-on question: Why not faavalancheFOAM, for example?

- I like how you distinguish between the 'mechanical model' on the one side and the 'process model' on the other side. Personally, I would refer to it as the kinematic description and the dynamic closure, but it essentially means the same thing. Since it is important to differentiate between the two, I suggest to have the content of the footnote on page 2 integrated into the main text. As a side remark: I had to read the footnote several times, in order to digest and understand it. I think readability would benefit from an additional comma between 'integration' and 'and'. Or maybe rephrase into two sentences. Generally, I think it would be good to explain the name faSavageHutteFOAM (if you want to stick to it) based on the difference between mechanical and process model. It seems to me that the 'mechanical model' solved in this paper is similar to the Savage-Hutter mechanical model, while the 'process model' had been altered.

- Section 2.1: In the interest of the reader, it would be good to work over the structure of section 2.1.. Two concrete suggestions:
  1) Include a brief explaination of equations (1)-(3) just after they have been introduced rather than stating: Variables and mathematical operators are explained later .... I find it OK to refer to Rauter and Tukovic (2018) for details, but it must be possible to understand the paper without. In that context: Is there is typo in the last term of equation (2), what's the 'e'?
  2) Why not taking up the earlier thought ('mechanical model' vs 'model closure') in the structure of section 2.1, e.g a subsection mechanical model and its solution (description of the numerics, etc.), and a subsection on model closures , e.g. equations (9) - (12).

- Section 2.1: page 6 equation (11). Is z here the actual altitude rather than vertical coordinate? If so please make this more explicit.

- Section 2.1: page 6 line 20-30: I like including a code snippet. Given that the finite area aspect is new to most readers it would be very instructive to see a code snippet that has either a tangential or a normal operator in it. Does OpenFOAM have different operators for this? Or do you specifically implement their definition into the code?

- Section 2.2: page 6 line 14: 'Numerical diffusivity' does not 'prevent' oscillations! Diffusive behavior is rather a feature of first-order methods, while oscillations and lower diffusivity is a feature of second order methods. Please adjust the sentence accordingly.

- The authors talk about the model to work in 'moderately curved' topographies several times. Is it possible to quantify this somehow? What curvature values can be dealt with and what values are critical and can't be dealt with? Or from the perspective of a potential user of the code: Can you put this rather fuzzy observation in a concrete guideline and provide information for which cases it is not possible to use the code? If this is not possible, at least lay out what would have to be done to find such thresholds?

- Section 2.3: I like the description of the opensource workflow. A simple question that I could imagine some readers (inlcuding me) would be interested in: Is it also possible to run the OpenFOAM Solver with any arbitrary unstructured polygonial surface mesh? If yes, in which format does it have to come? Can you comment on this?

- Section 4: What the authors refer to as numerical uncertainty seems to be established in Roache 1997. Please repeat its definition here in the paper as it is unclear. It should be simply a measure of the accuracy of the numerical method, right? In that case the important aspect is that it can be controlled by the scheme itself (whereas typically other sources of uncertainty can't be easily controlled). This is why I don't like the expression 'numerical uncertainty too much, personally). I do disagree with your statement on page 15 around line 5, that the numerical implementation influences the results dramatically. I rather see a tendency that carefully implemented and quality assessed numerical schemes of different type solving the same mathematical model get closer and closer. What do you mean here exactly?

- Section 4: I like the comparison of the OpenFOAM results with results of other codes. To me it would however seem very valuable to compare against a model that is more similar in spirit, e.g. the Voellmy Salm portion of the r.avaflow project for instance, simply because it is also implemented as a finite volume based method, and the sources for differences could be tracked down more easily. Any reasons, why you chose the samosAT path instead?

All the best, Julia

---

## Author Comment (AC3) · 23 May 2018

The manuscript faSavageHutterFOAM 1.0: Depth-integrated simulation of dense snow avalanches on natural terrain with OpenFOAM is a well-written contribution to the field of modeling and simulation of dense snow avalanches and fits into the scope of the journal. It extends the description of the OpenFOAM implementation first published by one of the authors, Matthias Rauter in (Rauter and Tukovic, 2018), to realistic topographies. The authors introduce the OpenFOAM specific workflow with a special focus on mesh generation from DEM data and GIS compatible post-processing. They demonstrate the capability of the proposed solver while analyzing a specific case

study, namely the Wolfsgruben avalanche. Simulation results of the new OpenFOAM based solver are compared to the established tool samosAT.

Dear Dr. Kowalski,

thank you very much for your detailed review. You uncovered some noteworthy aspects which we try to discuss in the following as well as in the revised manuscript.

My overall impression of this article is very positive due to the following reasons:

• First of all, by implementing the mathematical model into the mature software framework OpenFOAM, the authors actually outsource a lot of issues associated with e.g. data structure, parallelization and linear solvers. This allows a focus on the mathematical/physical model formulation itself, which is in my opinion the correct way to go.

-

• The mathematical/physical model itself is passed over to the solver in an encapsulated way as demonstrated on page 6 line 20 - 30. I find this a very big advantage of the approach. In fact this seems to be so straight forward, that one of the other reviewers even objected on the relative ease with which the model can be applied to real hazard scenarios, and the responsibility that arises from that. I certainly do have a slightly different opinion: While we as modelers do indeed have the responsibility to investigate and clearly communicate any limitations to the model, I fully support the authors approach in aiming towards modularizing software development and mathematical model representation as far as possible.

[Figure]

Thank you very much. Indeed, this was the main motivation of this work.

• Another aspect of the paper that I like very much is the comparison to an alternative simulation model. Although I have to admit that personally I find SamosAT a somewhat surprising choice (see my comment further down), this in principle is exactly the kind of results that help to better assess scope, potential and limitation of the various software tools that are in use.

-

• Finally, I want to emphasize the author's attempt to publish reproducible results. The paper refers to the code, which can be downloaded and tested. All in all, I am clearly in favor of publishing this paper and think it will be a valuable contribution to the field, if the following objections have been addressed.

-

• Title seems odd: Probably the title is chosen according to the OpenFOAM module - still I think it doesn't reflect the content of the paper. While the mechanical model that the authors solve is similar to the Savage-Hutter model (see also my next comment), some of the original Savage Hutter theory's defining flavors, e.g. its questionable active-passive earth-pressure term and the 'dry-friction-only' resistive force, are absent. This , however, is also OK, as it does not seem to be what the authors want to promote, compare and discuss in their paper. Straight-on question: Why not faavalancheFOAM, for example?

According to the journal policy, the main paper must give the model name and version number (or other unique identifier) in the title.

The OpenFOAM solver itself is called faSavageHutter to distinguish it from strictly two-dimensional ones with horizontal velocity (e.g. ShallowFOAM, https://github.com/mintgen/shallowFoam). The name is also owed to the historical development, as the classic Savage-Hutter model was our starting point. We kept the name because we stuck to the mechanical model of Savage & Hutter. It is true that we changed closure (or process) models to match the current practice. However, these changes do not influence solver code but solely added user-selectable submodels. Similarly, OpenFOAMs Navier-Stokes solver do not change their name when applying non-Newtonian viscosity models.

We will consider renaming the solver to 'faAvalancheFoam' in the future, however, the name of the solver in the current OpenFOAM release (OpenFOAM-v1712) can not be changed.

We will explain that the name refers to the mechanical description in the revised manuscript.

• I like how you distinguish between the 'mechanical model' on the one side and the 'process model' on the other side. Personally, I would refer to it as the kinematic description and the dynamic closure, but it essentially means the same thing. Since it is important to differentiate between the two, I suggest to have the content of the footnote on page 2 integrated into the main text. As a side remark: I had to read the footnote several times, in order to digest and understand it. I think readability would benefit from an additional comma between 'integration' and 'and'. Or maybe rephrase into two sentences. Generally, I think it would be good to explain the name faSavageHutteFOAM (if you want to stick to it) based on the difference between mechanical and process model. It seems to me that the 'mechanical model' solved in

this paper is similar to the Savage-Hutter mechanical model, while the 'process model' had been altered.

Distinguishing between 'mechanical model' and 'process model' or kinematic description and dynamic closure is common practice in the OpenFOAM community.

We will move the footnote into the text and rework phrasing to increase readability in the next revision.

• Section 2.1: In the interest of the reader, it would be good to work over the structure of section 2.1.. Two concrete suggestions: 1) Include a brief explaination of equations (1)-(3) just after they have been introduced rather than stating: Variables and mathematical operators are explained later .... I find it OK to refer to Rauter and Tukovic (2018) for details, but it must be possible to understand the paper without. In that context: Is there is typo in the last term of equation (2), what's the 'e'? 2) Why not taking up the earlier thought ('mechanical model' vs 'model closure') in the structure of section 2.1, e.g a subsection mechanical model and its solution (description of the numerics, etc.), and a subsection on model closures , e.g. equations (9) - (12).

We tried to follow the structure you suggest:
P. 4, line 5 to p. 5, line 16: kinematic description,
p. 5, line 17 to p. 6, line 12: dynamic closure,
p. 6, line 13 to p. 6, line 31: numerical solution and implementation.

We will add the respective sub-sub-sections entries in the revised manuscript:
2.1 Flow model,
2.1.1 Kinematic description,
2.1.2 Dynamic closure and
2.1.3 Numerical solution.

You're right about that equations are not explained properly, for readers not familiar with the classic Savage-Hutter model. We will explain individual terms in Eqs. (1) - (3), i.e. temporal derivative, advection, entrainment, basal friction and lateral pressure gradient. Moreover, we will add an introduction into surface projections to the appendix (see answers to reviewer 1 and 2).

The 'e' is indeed a typo, which we will remove in the revised version.

• Section 2.1: page 6 equation (11). Is $z$ here the actual altitude rather than vertical coordinate? If so please make this more explicit.

z is the actual altitude and it has to be consistent with the vertical coordinate in order for our preprocessing tool to work. We will make this more explicit in the revised manuscript.

• Section 2.1: page 6 line 20-30: I like including a code snippet. Given that the finite area aspect is new to most readers it would be very instructive to see a code snippet that has either a tangential or a normal operator in it. Does OpenFOAM have different operators for this? Or do you specifically implement their definition into the code?

Historically, the standard operators, `fam::grad(...)` and `fam:div(...)` contain the surface tangential projection if the respective result is a vector. The operators `fam::ngrad(...)` and `fam:ndiv(...)` have been added lately and contain the surface normal projection. All operators are mentioned in Rauter and Tukovic (2018) and we would prefer to refer to it in the revised manuscript.

• Section 2.2: page 6 line 14: 'Numerical diffusivity' does not 'prevent' oscillations!

Diffusive behavior is rather a feature of first-order methods, while oscillations and lower diffusivity is a feature of second order methods. Please adjust the sentence accordingly.

We understand that this paragraph is misleading and we will rework it in the revised manuscript.

• The authors talk about the model to work in 'moderately curved' topographies several times. Is it possible to quantify this somehow? What curvature values can be dealt with and what values are critical and can't be dealt with? Or from the perspective of a potential user of the code: Can you put this rather fuzzy observation in a concrete guideline and provide information for which cases it is not possible to use the code? If this is not possible, at least lay out what would have to be done to find such thresholds?

Greve et al. (1994) assumed a ratio between typical avalanche lenght L and curvature radius $R$ of order $O(\epsilon^{1/2})$, and a ratio between flow thickness $H$ and avalanche length of order $O(\epsilon)$, where $\epsilon \ll 1$. From this, we can deduce that $H/R = O(\epsilon^{1/2})$, i.e. that flow height is significantly smaller than the curvature radius. This allows depth-integration in surface normal direction. In our experience, this assumption is often violated in practice. At the moment, it is not possible for us to quantify the effects of strong curvature on results. A corresponding investigation would certainly be appropriate, but would go beyond the scope of this work. We will therefore explain what 'mildly curved' means and add a comment on strong curvature in the outlook of the revised manuscript.

• Section 2.3: I like the description of the opensource workflow. A simple question that I could imagine some readers (inlcuding me) would be interested in: Is it

also possible to run the OpenFOAM Solver with any arbitrary unstructured polygonial surface mesh? If yes, in which format does it have to come? Can you comment on this?

The mesh should be a valid Finite Volume / Finite Area Mesh, which enforces some constraints, e.g. mostly flat and convex faces.

Moreover, the solver expects the mesh in OpenFOAM format. However, conversion tools for many formats are available (see https://cfd.direct/openfoam/user-guide/standard-utilities/, section 3.6.3, Mesh conversion). Moreover, a custom conversion tool can be written using the OpenFOAM C++ library (see, e.g. supplementary material, file slopeMesh/slopeMesh.C, lines 307-362, where a new mesh is created with known edge points and connections). Finally, the OpenFOAM mesh format consists of ascii-files and can therefore be easily written with custom tools (see, e.g. supplementary material, file scripts/txt2mesh.py, lines 472-670). Note that the Finite Area Mesh file format just contains references to Finite Volume Mesh files, meaning that a Finite Volume Mesh has to be present. However, this can be handled by creating a dummy FVM. We will add a hint to mesh conversion tools in the revised manuscript.

• Section 4: What the authors refer to as numerical uncertainty seems to be established in Roache 1997. Please repeat its definition here in the paper as it is unclear. It should be simply a measure of the accuracy of the numerical method, right? In that case the important aspect is that it can be controlled by the scheme itself (whereas typically other sources of uncertainty can't be easily controlled). This is why I don't like the expression 'numerical uncertainty too much, personally). I do disagree with your statement on page 15 around line 5, that the numerical implementation influences the results dramatically. I rather see a tendency that carefully implemented and quality assessed numerical schemes of different type solving the same mathematical model get closer and closer. What do you mean here exactly?

[Figure]

The uncertainty estimation following Roache is indeed a measure of the accuracy and an estimation of the related error, based on a Richardson Extrapolation. This is mentioned in the method section, page 7, line 25.

Sod (1978) showed high differences between different methods when solving the compressible Navier-Stokes equations. Ferziger and Peric (2002), for example, report that there are substantial differences in results, even when using the same method/model: "Recent comparative studies in which the same problem was solved by different groups using different codes reveal that the differences between solutions obtained using different codes with the same models are often larger than the differences between solutions obtained using the same code and different models (Bradshaw et al., 1994; Rodi et al., 1995). These differences can only be due to numerical error or user mistakes, if the models are really the same (it is not unusual that supposedly same models turn out to be different due to different interpretation, implementation, or boundary treatment)." Therefore, we think that differences between SamosAT and our code have to be expected. However, we see that referring to Sod (1987) may be misleading at this point and we will change this reference to Ferziger and Peric (2002) in the revised manuscript.

• Section 4: I like the comparison of the OpenFOAM results with results of other codes. To me it would however seem very valuable to compare against a model that is more similar in spirit, e.g. the Voellmy Salm portion of the r.avaflow project for instance, simply because it is also implemented as a finite volume based method, and the sources for differences could be tracked down more easily. Any reasons, why you chose the samosAT path instead?

We chose SamosAT because we trust its method (Lagrangian; SPH) in terms of

complex topography (see manuscript, page 7, lines 7-9) and because we have experience using it (Fischer et al. 2015; Rauter et al. 2016). In particular, r.avaflow, as described by Mergili et al. (2012), seems to be limited in terms of complex topography, since it is using the model of Gray et al. (1999).

---

## Author Response (AR1)

**Final answer regarding "faSavageHutterFOAM 1.0: Depth-integrated simulation of dense snow avalanches on natural terrain with OpenFOAM"**

The main suggestions for improvement of all three referees were related to the description of the model. We adapted the manuscript by extending the respective explanations (**pages 5-6, lines 21++**) and added an appendix to clarify the use of projections in surface partial differential equations (**pages 19-21**).

**Referee 1**

We addressed all comments by referee 1 in the revised manuscript to the best of our ability. In particular, we focused on improving the comprehensibility of model equations by clarifying and extending the respective explanations. While we share the reviewers concerns regarding the friction and entrainment models, we hope we gave a satisfying explanation as to why these topics are out of the scope of this paper.

**Specific changes following referee comments:**

**1) Entrainment term in momentum conservation equation:**
**page 5, lines 23-24:** We state that the conservative form shows no entrainment term in the momentum conservation equation and that this may be different for the non-conservative form.

**2-4) Validity of entrainment model**
We agree with the referee that the entrainment model should be improved and validated carefully. However, this is out of the scope of this paper.

**5) Stopping of the avalanche**
**page 7, lines 2-4:** We explained how the method allows the avalanche to stop and the effect of the parameter $u_0$. **page 6, line 27:** We report the value chosen for the parameter $u_0$. **page 14, line 2:** We explain that the creeping velocity is lower than the solver tolerance.

**6) Name of the Solver**
**page 3, lines 18-19:** The solver name is explained in the introduction now. For the discussed reasons, we can not change the solver name but will consider it in the future.

**7-10) Model equations**
**page 3, lines 18-19:** The discussion of model equations has been extended. **pages 19-21:** The projections are now explained in depth in appendix A.

**11) Validity of friction model**
We agree with the reviewer that the classic Voellmy friction model shows some issues. However, it is widely used in practice and therefore fitting into this work, where no focus lies on the friction models.

**12) Are oscillation suppressed by numerical schemes?**
We answered this question in the answer to the review.

**13) Scales in figures are missing.**
We added missing scales and labels where the software allowed us to do so, however, Paraview is limited in this regard. **page 16, figure 7:** Labels have been added to contour lines.

**14) Modulus signs are missing.**
**page 14, line 3:** Modulus signs have been added.

**15) Comparisons of simulation to deposition**
We agree with the referee that more comparisons would be appropriate, however this is out of the scope of this paper for reasons discussed in the answer to the review.

**Referee 2**

We addressed all remarks by referee 2 in the revised manuscript. Mainly we worked on clarifying the explanation of the model equations, as suggested by the reviewer. .

**Specific changes following referee comments:**

**1-2) Understanding of model equations.**
**pages 5-6, lines 21++:** Explanation has been extended. **pages 19-21:** The projections are now explained in depth in appendix A.

**3) Effect of $u_0$**
**page 6, line 27:** The chosen value for $u_0$ is reported now. **pages 7, lines 2-4:** The effect of the parameter $u_0$ is explained in detail.

**4) Regularisation of entrainment**
**page 7, lines 17-18:** We report how the regularisation of entrainment is implemented.

**Referee 3**

We addressed all remarks by referee 2 in the revised manuscript. Mainly we worked on clarifying the explanation of the model equations, as suggested by the reviewer. .

**Specific changes following referee comments:**

**1-5) Comments on paper concept, open source, data availability**
Nothing to address.

**6) Title seems odd**
**page 3, lines 18-19:** The solver name is explained in the introduction now. For the discussed reasons, we can not change the solver name but will consider it in the future.

**7) Distinction mechanical model - process model**
**page 2, lines 14-17:** The footnote has been moved into the text and split into multiple sentences to increase readability.

**8) Structure of model section**
**page 5, line 1; page 6, line 21; page 7, line 19:** Introduced sub-sub-sections 2.1.1, 2.1.2, 2.1.3.
**pages 5-6, lines 21++:** Explanation extended.
**page 5, line 6:** Removed typo in equation.

**9) Elevation vs. coordinate**

**page 7, line 15:** Made clear that $z$ is the altitude and in our case also the coordinate.

**10) Code snippet with tangential or a normal operator**

We stick to the simple scalar operator because others are too complex to show them without extensive explanation. However, on **page 7, line 15**, we refer to Rauter and Tukovic (2018), where all code of all equations is listed.

**11) 'Mildly curved'**

**page 2, lines 1-2:** We state what we mean with mildly curved terrain. **page 19, lines 17-18:** We state that the limitation to mildly curved terrain should be eliminated in the future.

**12) Meshing tools**

**page 9, lines 16-17:** We added a note with a hint on mesh conversion tools.

**13) Numerical uncertainty**

**page 15, lines 18-19:** Misleading text changed according to referee comment.

**14) Choice of software for comparison**

We explained the choice for SamosAT in author answers.

[revised manuscript text omitted]